# Unexpected contribution of fibroblasts to muscle lineage as a mechanism for limb muscle patterning

Joana Esteves de Lima [1,2,3,9], Cédrine Blavet[1,2,9], Marie-Ange Bonnin[1,2,9], Estelle Hirsinger [1,2,9], Glenda Comai [4], Laurent Yvernogeau[1,2,5], Marie-Claire Delfini [1,2,6], Léa Bellenger [7], Sébastien Mella[4], Sonya Nassari[1,2], Catherine Robin [5], Ronen Schweitzer[8], Claire Fournier-Thibault[1,2], Thierry Jaffredo [1,2], Shahragim Tajbakhsh [4], Frédéric Relaix [3] & Delphine Duprez [1,2✉]

Positional information driving limb muscle patterning is contained in connective tissue fibroblasts but not in myogenic cells. Limb muscles originate from somites, while connective tissues originate from lateral plate mesoderm. With cell and genetic lineage tracing we challenge this model and identify an unexpected contribution of lateral plate-derived fibroblasts to the myogenic lineage, preferentially at the myotendinous junction. Analysis of single-cell RNA-sequencing data from whole limbs at successive developmental stages identifies a population displaying a dual muscle and connective tissue signature. BMP signalling is active in this dual population and at the tendon/muscle interface. In vivo and in vitro gain- and loss-of-function experiments show that BMP signalling regulates a fibroblast-to-myoblast conversion. These results suggest a scenario in which BMP signalling converts a subset of lateral plate mesoderm-derived cells to a myogenic fate in order to create a boundary of fibroblast-derived myonuclei at the myotendinous junction that controls limb muscle patterning.

[1] Developmental Biology Laboratory, Institut Biologie Paris Seine, Sorbonne Université, CNRS, IBPS-UMR 7622, Paris, France. [2] Inserm U1156, Paris, France. [3] Univ Paris Est Creteil, INSERM, EnvA, EFS, AP-HP, IMRB, Creteil, France. [4] Department of Developmental and Stem Cell Biology, Institut Pasteur, CNRS UMR 3738, Paris, France. [5] Hubrecht Institute-Royal Netherlands Academy of Arts and Sciences (KNAW), Regenerative Medicine Center, University Medical Center Utrecht, Utrecht, The Netherlands. [6] Aix Marseille University, CNRS, IBDM, Marseille, France. [7] Institut Biologie Paris Seine, Sorbonne Université, CNRS, IBPS-FR3631, ARTbio Bioinformatics Platform, Inserm US 037, Paris, France. [8] Research Division, Shriners Hospital for Children, Portland, OR, USA. [9] These authors contributed equally: Joana Esteves de Lima, Cédrine Blavet, Marie-Ange Bonnin, Estelle Hirsinger. ✉email: Delphine.duprez@upmc.fr

Skeletal muscle patterning is a developmental process that controls the formation of each muscle at the right position and time, leading to the highly complex organisation of over 600 individual skeletal muscles in human, each of them with a specific shape, size, innervation and attachment sites to the skeletal system. The cellular and molecular mechanisms that regulate limb muscle patterning remain poorly understood, although we know from embryological experiments that the positional information is not contained in myogenic cells themselves but in connective tissues (CTs)[1–7]. CTs, primarily composed of fibroblasts and extracellular matrix, support and connect organs together[2]. Muscle attachments are composed of two types of CTs, the tendons that link muscle to bone and the muscle CT that is a continuum of tendon surrounding individual muscles and also present in between muscle fibres[2]. The long-standing consensus emerging from experimental embryology in avians is that limb myogenic cells are derived from somitic mesoderm, while CT fibroblasts originate from the lateral plate mesoderm[1,2,8–10]. Consistently, genetic lineage tracing experiments in mice during development have shown that PAX7+ muscle progenitors originate from the *Pax3* lineage in the limb[11–13] and that limb muscle attachment originates from CT fibroblast lineages[14,15]. Previous embryological surgical experiments in chicken embryos demonstrated that the positional information for limb muscles and associated innervation (motoneuron axons) is contained in the lateral plate mesoderm and not in somite-derived tissues[1,5,6,16–18]. This positional information for muscle patterning is maintained over time in lateral plate-derived tissues, such as muscle CT[3] and tendons[4]. The molecular signals produced by lateral plate-derived tissues that drive the correct positioning of limb muscles during development are not fully identified. To date, the TBX4/5, TCF4 and OSR1 transcription factors expressed in limb CT fibroblasts are recognised to act in a non-cell-autonomous manner to regulate limb muscle patterning[15,19,20]. While it is possible that CT fibroblasts and myogenic cells interact through secreted signals at the CT/muscle interface, the precise nature and mechanisms of these interactions are currently unknown. The CT/muscle interface is likely to be the place where muscle patterning occurs and one would expect that these junctional cells show specificities in relation to their interface status. The myotendinous junction is a type of CT/muscle interface, where the muscle attaches to the tendon, and it is fully formed at postnatal stages[21]. The mechanisms driving the establishment of the myotendinous junction during development are poorly studied. There is regionalisation of patterning signals at the tendon/muscle interface, since known signalling molecules display a restricted expression in a subset of myonuclei at muscle tips close to tendons during foetal myogenesis[22–25]. However, the function of these genes with regionalised expression in muscle patterning and myotendinous junction formation is not understood.

Here we identify an unexpected contribution of CT fibroblast nuclei to myotubes preferentially at the tips of muscles close to tendons, which provides a ~~novel~~ cellular mechanism underlying limb muscle patterning and subsequent myotendinous junction formation. We also found that BMP signalling regulates the recruitment of CT fibroblast nuclei to myotubes.

## Results

### A junctional population of muscle cells does not originate from somite.
To characterise the CT/muscle interface in developing limbs, we first revisited the embryological origins of CT and myogenic cells using cell lineage experiments in avian embryos and genetic lineage tracing in mice. As previously demonstrated[8,9], isotopic/isochronic quail-into-chicken presomitic mesoderm grafts performed at the presumptive limb regions showed that the vast majority of muscle cells were labelled with the quail-specific antibody, QCPN (Fig. 1a–c). This key experiment is at the basis of the dogma saying that limb muscle cells derive from somites. However, careful examination of these grafts also identified a subpopulation of myonuclei (nuclei of multinucleated myotubes) that were QCPN-negative and thus were not of somitic origin (Fig. 1d). At E9, when muscle patterning is set, these QCPN-negative myonuclei were located at the extremities of myosin+ myotubes, close to tendons visualised with collagen XII expression (Fig. 1a–e and Supplementary Fig. 1). Accordingly, a subset of MYOG+ myoblasts located at muscle tips was also not of somitic origin (QCPN-negative) (Fig. 1f, g). Quantification of MYOG+/QCPN− cells versus total MYOG+ cells shows that about 9.5% (±3.7 SD) of MYOG+ cells were not somite-derived (Fig. 1h). As the QCPN/PAX7 double immunolabelling could not be performed due to antibody incompatibility, we performed presomitic mesoderm grafts using the available cytoplasmic GFP-chicken reporter line[26]. This analysis identified a population of PAX7+GFP− cells at muscle tips in close proximity to tendons (Supplementary Fig. 2). From these presomitic mesoderm grafting experiments, we conclude that a small fraction of PAX7+ muscle progenitors, MYOG+ myoblasts and myonuclei are not derived from somites.

To determine the embryonic origin of these non-somite-derived muscle cells, we performed the converse experiment and replaced the limb lateral plate mesoderm of chicken embryos by the equivalent region of quail embryos. Consistent with classical lineage descriptions[1,8,9], this type of graft confirmed the lateral plate mesoderm origin of cartilage, tendons and muscle connective tissue (Supplementary Fig. 3). However, we also found lateral plate-derived myonuclei (QCPN+) at the extremities of myosin+ myotubes in close vicinity to tendons visualised with *SCX* expression at E9 (Fig. 1i–l). To assess lateral plate contribution to the muscle lineage, we electroporated the lateral plate mesoderm of chicken embryos at the forelimb level with a plasmid co-expressing membrane-Tomato and nuclear-GFP reporters under the control of a ubiquitous promoter[27]. Successfully electroporated forelimbs were screened for Tomato expression (Supplementary Fig. 4a–c) and nuclear-GFP labelling was used to accurately assess the contribution of lateral plate cells to syncytial myofibers and interstitial myogenic cells. As expected, GFP+ nuclei encompassed all lateral plate-derived tissues but in a mosaic manner (Supplementary Fig. 4d, e), unlike tissue graft experiments (Supplementary Fig. 3). Despite this mosaic labelling, we observed sporadic GFP+ myonuclei, GFP+MYOG+ myoblasts and GFP+PAX7+ muscle progenitors located at muscle tips in E10 chicken limbs (Supplementary Fig. 4f–i).

Altogether, these grafting and electroporation experiments performed in avian embryos show an unexpected contribution of lateral plate-derived cells to the myogenic lineage in limb muscles with a preferential location at the muscle tips close to tendons.

### *Scx* and *Osr1* lineages contribute to myogenic cells.
It is well recognised that the *Pax3* lineage corresponds to the somite-derivatives in mouse limbs[28]. To assess if the lateral plate contribution to limb muscles observed in chicken was conserved in mice, we analysed PAX7 expression with respect to the *Pax3* lineage. Using a Tomato reporter mouse line to trace the *Pax3* lineage[28,29], we observed the expected contribution of *Pax3*-derived cells to limb foetal muscles at E14.5 and E15.5 when the muscle pattern is set (Fig. 2a, b and Supplementary Fig. 5a). Notably, we observed a fraction of PAX7+ cells that were not *Pax3*-derived (Tomato-negative cells), with a preferential location close to tendons at foetal stages (Fig. 2a–d and Supplementary Fig. 5a–d). Quantifications showed that 4.6% (±1.35 SD)

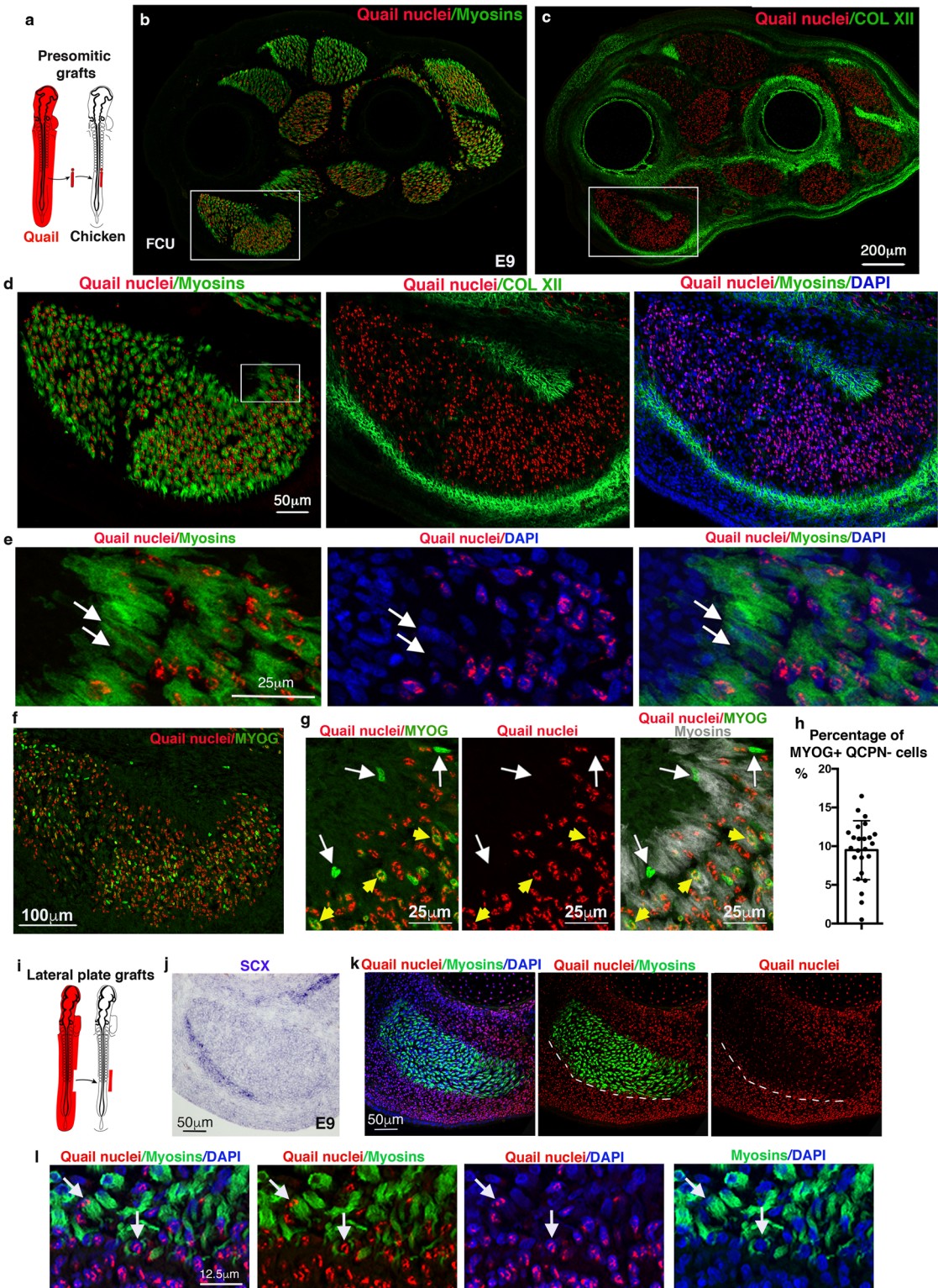

of PAX7+ cells were Tomato-negative at E15.5 (Fig. 2d), while 7.5% (±2.3 SD) of PAX7+ cells were Tomato-negative at E14.5 (Supplementary Fig. 5d). In addition, using a nuclear H2B-GFP reporter mouse line to trace the *Pax3* lineage[28,30], we further identified 3.1% (±1.1 SD) of PAX7+ muscle progenitors and 2.6% (±1.2 SD) of MYOD+ myoblasts that were not *Pax3*-derived (nuclear-GFP-negative cells), at muscle tips close to tendons at E15.5 (Fig. 2e–h). We conclude that a small fraction of PAX7+ and MYOD+ myogenic cells are not derived from the

*Pax3* lineage in mouse limb muscles at the foetal stage. These results are in contrast to the established model supporting that *Pax3*+ somitic cells give rise to the entirety of PAX7+ cells in mouse limbs[11–13].

To determine the developmental origin of these non-*Pax3*-derived myogenic cells, we performed genetic lineage tracing analyses with the CT fibroblast markers, *Scx* and *Osr1*.

The bHLH transcription factor SCX is a recognised tendon cell marker during development and is involved in tendon

**Fig. 1 A subpopulation of muscle cells does not originate from somites but from the lateral plate in the chicken limb. a–c** Limb muscle organisation after quail-into-chicken presomitic mesoderm grafts. **a** Schematic of limb presomitic mesoderm grafts. **b**, **c** Transverse and adjacent forelimb sections of E9 presomitic mesoderm grafted embryos immunostained with QCPN (quail nuclei, red) and MF20 (myosins, green) (**b**), and with QCPN (quail nuclei, red) and ColXII (tendons, green) antibodies (**c**). **d** High magnification of FCU, a ventral and posterior muscle immunostained with QCPN (quail nuclei, red), MF20 (myosins, green) and COLXII (tendons, green) antibodies combined with DAPI (nuclei, blue). **e** High magnification of muscle tips (close to tendons), squared in **d**, showing non-quail nuclei within myosin+ cells (arrows). **f** High magnification of FCU immunostained with QCPN (quail nuclei, red) and MYOG (green) antibodies. **g** Focus on muscle tip regions showing MYOG+ nuclei (green) that are not quail+ (red) (white arrows) close to myosin+ cells (grey). Yellow arrowheads point to MYOG+ (green) that are quail+ (red). **h** Percentage of MYOG+/QCPN− cells versus total MYOG+ cells. Data are presented as mean values ± SD from 3 grafted limbs. **i–l** Limb muscle organisation after quail-into-chicken lateral plate mesoderm grafts. **i** Schematic of limb lateral plate mesoderm grafts. **j**, **k** Transverse and adjacent limb sections were hybridised with SCX probe (blue) to label tendons (**j**) and immunostained with QCPN (quail nuclei, red), MF20 (myosins, green) antibodies combined with DAPI (blue) (**k**), focused on the FCU muscle. **l** High magnification of muscle tip regions (close to tendons) showing the high density of quail nuclei of lateral plate mesoderm origin. Arrows point to quail myonuclei (red) in myosin+ fibres (green).

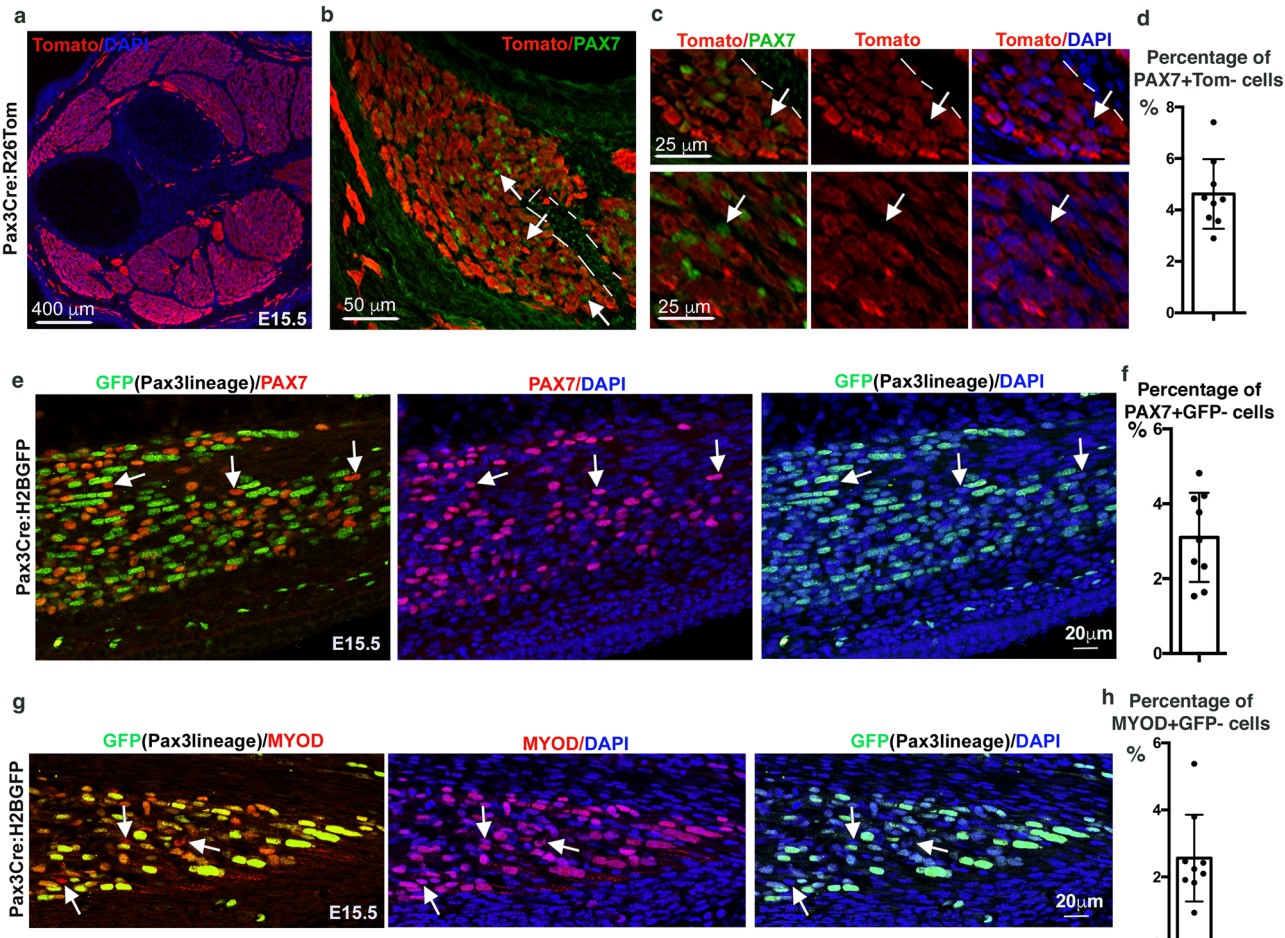

**Fig. 2 A subpopulation of muscle cells is not derived from the *Pax3* lineage in mouse limbs. a** Transverse forelimb sections of E15.5 *Pax3^{Cre}:R26^{stop/Tom}* mice immunostained with Tomato (*Pax3* lineage, red) antibody and DAPI staining (blue). **b** Focus on a muscle immunostained with Tomato (*Pax3* lineage, red) and PAX7 (green) antibodies. **c** High magnifications showing PAX7+ cells (green) that are tomato-negative, i.e. not of *Pax3* lineage (arrows). **d** Percentage of PAX7+/Tomato− cells versus PAX7+ cells in muscles. Data are presented as mean values ± SD from 3 limbs. **e** Longitudinal sections along a limb muscle of E15.5 *Pax3^{Cre}:R26^{H2BGFP}* mice immunostained with GFP (*Pax3* lineage, green) and PAX7 (red) antibodies. Arrows point to PAX7+ cells that are GFP-negative. **f** Percentage of PAX7+/GFP− cells versus PAX7+ cells in muscles. Data are presented as mean values ± SD from 3 limbs. **g** Longitudinal sections along a limb muscle of E15.5 *Pax3^{Cre}:R26^{H2BGFP}* mice immunostained with GFP (*Pax3* lineage, green) and MYOD (red) antibodies. Arrows point to MYOD+ cells that are GFP-negative. **h** Percentage of MYOD+/GFP− cells versus MYOD+ cells in E15.5 muscles. Data are presented as mean values ± SD from 3 limbs.

formation[31,32]. We used the *Scx^{Cre}:R26^{stop/Tom}:Scx-GFP* mice[14,29] to analyse contemporary *Scx* expression together with the *Scx* lineage contribution in developing limbs. As described previously[14], the *Scx* lineage (Tomato+ cells) was observed in the limb mesenchyme with a pattern overlapping that of the *Scx-*

GFP transgene in tendons (Fig. 3a). At E16.5, an overlap between myosins and Tomato labelling was observed in Tomato+ junctional regions between muscles (Supplementary Fig. 5e, f). In addition to tendon cells, we identified PAX7+ and MYOD/ MYOG+ myogenic cells within the *Scx* lineage that were

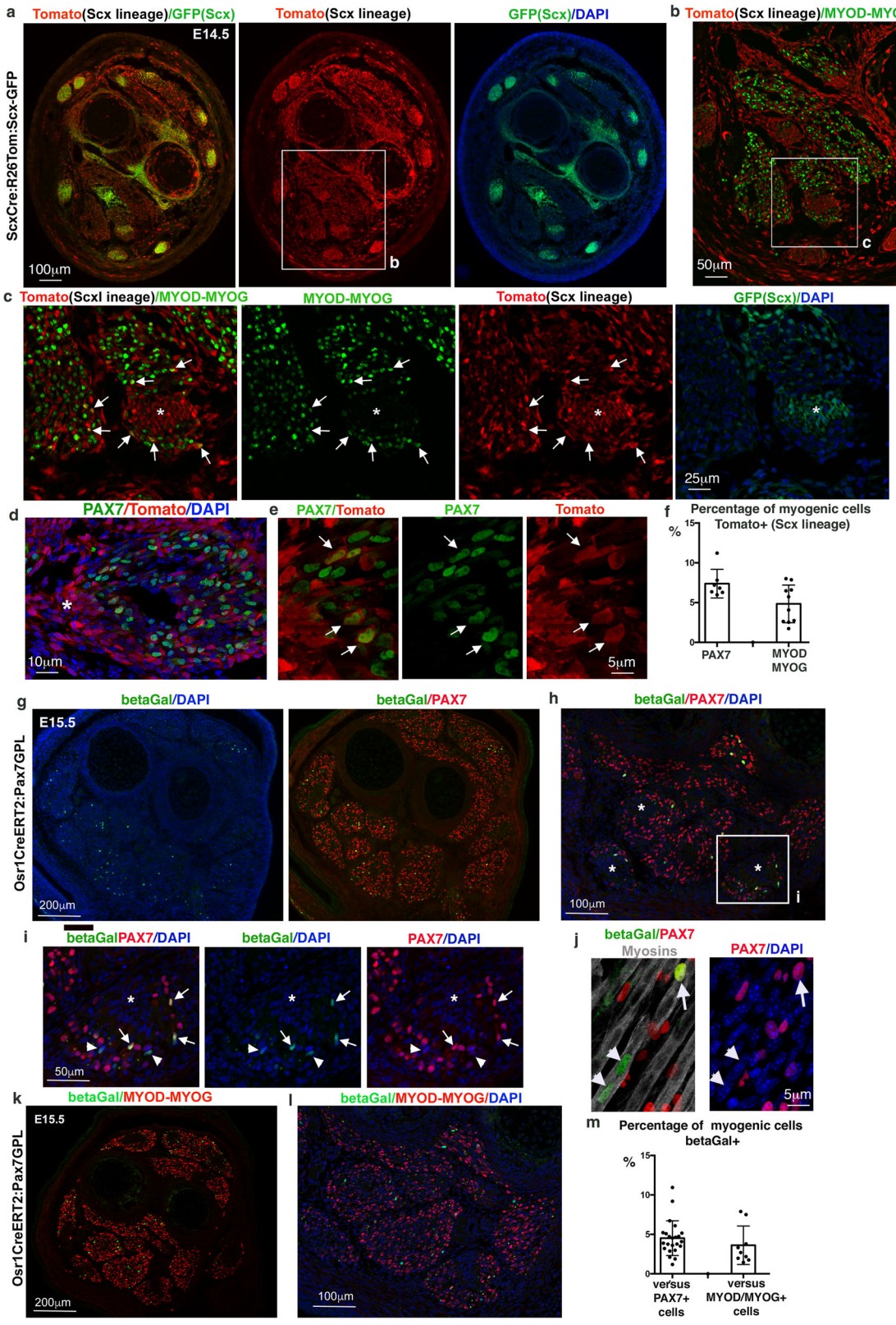

preferentially concentrated close to tendons at E14.5 (Fig. 3b–e). Quantifications showed that 7.4% (±1.7 SD) of PAX7+ cells and 4.8% (±2.2 SD) of MYOD/MYOG+ cells were derived from *Scx* lineage (Tomato+ cells) (Fig. 3f). At E12.5, when the muscles are not yet individualised, we found 4.12% (±1.3 SD) of PAX7+ and 4.13% (±1.15 SD) of MYOD/MYOG+ myogenic cells surrounding the muscle masses to be *Scx*-derived (Tomato+)

(Supplementary Fig. 5g–k). These values are in accordance with those obtained from the *Pax3* lineage analysis.

The zinc-finger transcription factor OSR1 is a key regulator of muscle CT differentiation[15]. As already reported[15,33], lineage tracing using *Osr1CreERT2* as a driver[15,33] with a ubiquitous Tomato reporter[29] led to Tomato expression in limb muscle CT around individual muscles and in between myotubes

**Fig. 3 Scx and Osr1 lineages provide myogenic cells in foetal mouse limbs. a–c** Transverse forelimb sections of E14.5 $Scx^{Cre}:Rosa26^{Tom}:Scx^{(GFP)}$ mice. **a** Tomato expression shows *Scx* lineage in tendons and limb CTs (red), while GFP shows *Scx* expression in tendons (green). **b** Focus on ventral muscles labelled with both MYOD and MYOG antibodies (green). **c** Focus on myogenic cells MYOD/MYOG+ cells that are Tomato+ (red, arrows), close to a tendon labelled with GFP(Scx) (white asterisks). **d, e** Longitudinal sections along a limb muscle of E14.5 $Scx^{Cre}:Rosa26^{Tom}:Scx^{(GFP)}$ mice labelled with PAX7 (green) and Tomato (red) antibodies. Tendon is highlighted with a white asterisk. Panel **e** is a high magnification of muscle tips. Arrows point to PAX7+ cells (green) that are Tomato+ (red, *Scx* lineage), close to the tendon (white asterisk). **f** Percentage of PAX7+/Tomato+ cells versus PAX7+ cells and percentage of MYOD-MYOG+/Tomato+ cells versus MYOD/MYOG+ cells. Data are presented as mean values ± SD from 3 limbs. **g–j** Forelimb sections of E15.5 $Osr1^{CreERT2}:Pax7^{GPL}$ mice (with tamoxifen injection at E11.5) immunostained with betaGal (*Pax7* lineage in *Osr1*-lineage-derived cells, green) and PAX7 (red) antibodies, combined with DAPI (blue). **g** Global view of a forelimb transverse section. **h** Focus on ventral limb muscles. Tendons are highlighted with white asterisks. **i** Focus on a muscle region close to a tendon (asterisks). Arrows point to betaGal+ cells (green) that are PAX7+ (red), while arrowheads point to betaGal+ cells that are not PAX7+ anymore. **j** High magnification of betaGal+ cells (green), arrows point to the betaGal +/PAX7+ nuclei that are located in between myotubes (myosins, grey), while arrowheads point to the betaGal+ nuclei (green) in which PAX7 has been downregulated after cell incorporation into myotubes. **k, l** Forelimb sections of E15.5 $Osr1^{CreERT2}:Pax7^{GPL}$ mice immunostained with betaGal (green) and MYOD-MYOG (red) antibodies, combined with DAPI (blue). **l** is a focus of ventral limb muscles of **k**. **m** Percentage of betaGal+ versus PAX7+ cells and versus MYOD-MYOG+ cells in limb muscles of E15.5 embryos. Data are presented as mean values ± SD from 3 limbs.

(Supplementary Fig. 6a, b). To address the *Osr1* lineage contribution to the muscle lineage specifically, we performed lineage tracing with $Osr1^{CreERT2}$ as a driver and the *Pax7* reporter ($Pax7^{GFP-Puro-nlacZ}$, hereafter $Pax7^{GPL}$), a reporter gene that is not expressed in CT cells[34]. As such, only cells that expressed CreERT2 from the *Osr1* locus and belong to the myogenic lineage (*Pax7*+ and descendants) will express nuclear β-galactosidase (betaGal) upon Tamoxifen induction[34]. Using this genetic combination, we found 4.5% (±2.5 SD) of PAX7+ muscle progenitors to be of *Osr1* lineage (betaGal+ cells) at E15.5 upon tamoxifen induction at E10.5 (Fig. 3g–j, m). Similarly, 3.6% (±2.4 SD) of MYOD/MYOG+ myogenic cells were derived from *Osr1* lineage (betaGal+ cells) (Fig. 3k–m). A slightly higher contribution of the *Osr1* lineage to PAX7+ cells (6.2% ± 2.8 SD) and MYOD/MYOG+ cells (6.9% ± 3 SD) was noted at an earlier time point (E12.5) in limb muscle masses (Supplementary Fig. 6c–j). To define the time window of the contribution of *Osr1* lineage to myogenic cells, we performed the tamoxifen injection at later time points. We observed only sparse (induction at E12.5) or no (induction at E13.5) betaGal+ cells in foetal limb muscles (Supplementary Fig. 6k–m). Finally, we combined the $Osr1^{CreERT2}$ line with the conditional Fucci2aR[35] reporter, where mCherry labels nuclei in the G1 phase of the cell cycle[36] as another nuclear reporter system to label the post-mitotic myonuclei. When tamoxifen injections were performed at an earlier time point (induction at E8.5), we found abundant Fucci+ myonuclei (*Osr1*-lineage origin) in myotubes (Supplementary Fig. 6n, o). Altogether, we conclude that as for the *Scx* lineage, the *Osr1* lineage contributes to myogenic cells and this occurs mainly before E12.5, at the stage where important spatial arrangements of muscle and CT organisation take place in the developing limb.

Given that *Pax7* is expressed in a subset of neural crest cells in chicken limbs, we verified the absence of neural crest cell contribution to the muscle lineage using isotopic/isochronic GFP +chicken-into-chicken grafts of neural tubes and genetic lineage tracing experiments in limbs of $Wnt1^{Cre}$;$R26^{stop}$/Tom mice[37]. Using these approaches, we did not find a contribution of neural crest cells to either myogenic cells or myonuclei at muscle tips close to tendons in chicken and mouse limbs (Supplementary Fig. 7).

Altogether, the cell lineage tracing experiments in avians and mice led us to conclude that a small fraction (4–9%) of myogenic cells, preferentially located at muscle tips close to tendons, are derived from limb CT lineages and not from the *Pax3* somitic lineages.

**scRNAseq analysis identifies dual CT/myogenic cells in limbs.** The unexpected contribution of CT fibroblasts to the muscle

lineage as early as the PAX7+ muscle progenitor stage suggested a scenario in which CT fibroblasts acquire progressively a myogenic signature to be incorporated at the tips of existing myotubes during development. In this scenario, CT fibroblasts would convert to a myogenic fate before fusion. To search for cells exhibiting a transient CT/muscle signature, we turned to the global transcriptomic analysis of limb cells with single-cell resolution by single-cell RNA-sequencing (scRNAseq) of chicken whole-limb cells at successive developmental stages: E4, a progenitor stage; E6, when major spatial re-arrangements occur for muscle and CTs and E10, when the final muscle pattern is set. At each stage, we identified CT and muscle clusters, along with their respective CT and muscle markers (Fig. 4a and Supplementary Table 1). In parallel, we identified a subpopulation of cells co-expressing, at the cellular level, at least one CT-marker (among PRRX1[38], PDGFRA[39], TWIST2[40], OSR1[15] and SCX[31]) and one muscle marker (among PAX7, MYF5, MYOD and MYOG) (Supplementary Fig. 8a). The size of this dual CT/muscle (CT/M) population increased from 0.5% of limb cells at E4 to reach 4% of limb cells at E6 and was maintained at 3.3% at E10 (Fig. 4b). At E10, 11% of the MYOG+ myoblasts had a dual phenotype signature, which is consistent with the 9.5% of MYOG+ myoblasts that were not somite-derived in our quail-chick grafting experiments (Fig. 1h). We found that quantifications from the scRNAseq analysis were lower compared to those of the mouse genetic lineage tracing experiments. For example, at E10, 1.6% of PAX7+ cells were PAX7+/OSR1+ dual cells compared to 4.5% in the $Osr1^{CreERT2}:Pax7^{GFP}$ mouse embryos at E15.5 (Fig. 3m). This was expected as in the latter, the analysis is based on lineage history whereas contemporary expression was examined in the transcriptome analysis. CT and M scores were calculated as the corrected average expression levels of the CT and muscle markers, respectively (Supplementary Table 1) and plotted for each cell grouped by its CT, CT/M or M identity (Fig. 4c). Except at E4 when the CT score was the same for the CT and CT/M populations, the CT/M cells exhibited intermediate CT and M scores, compared to those of cells with a strict CT or muscle identity (Fig. 4c), pointing to a hybrid CT/muscle identity. The analysis of genes differentially expressed in the CT/M population versus the combined CT and muscle populations showed that these CT/M cells did not express specific genes other than CT markers or muscle markers (Supplementary Table 2, gene examples in Fig. 4d). As a consequence, the CT/M population does not cluster but rather is included in the CT and muscle clusters (Fig. 4e). CT/M cells with a high CT score are found within CT clusters, while those with a high M score are found within the muscle clusters (Supplementary Fig. 8b). Another interesting feature is that CT markers were systematically

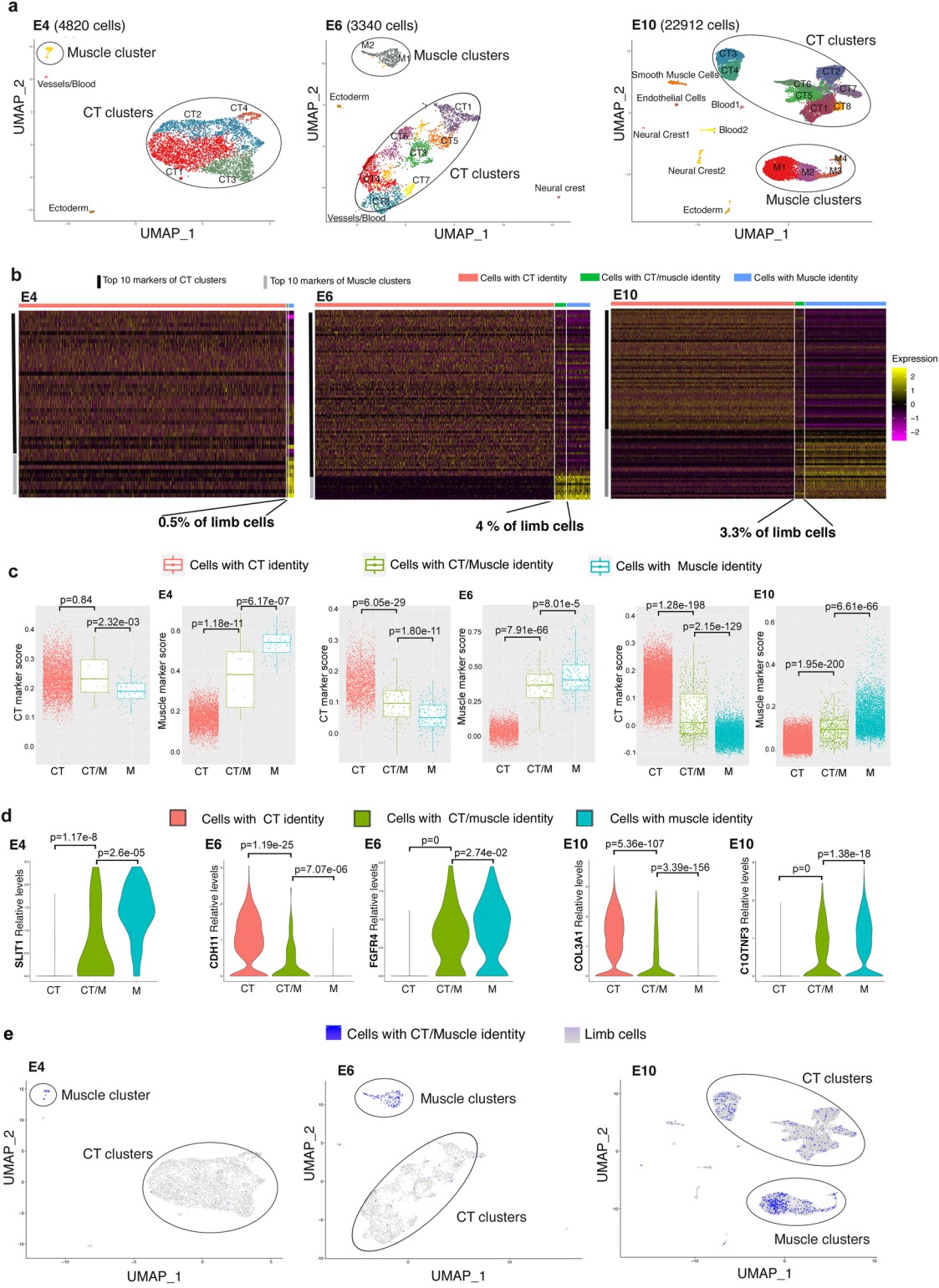

downregulated (except at E4) and muscle markers were systematically upregulated in the CT/M cells (Supplementary Table 2). This bi-phasic expression profile suggests that the CT/M cells were in transit from a CT identity to a muscle identity. At E4, CT markers are not yet downregulated and M markers are already upregulated in CT/M cells (Supplementary Table 2), suggesting that the transition process starts before E6 by the upregulation of muscle markers followed by a subsequent downregulation of CT markers in the CT/M cells.

Among CT markers, *PRRX1* was expressed in the majority of CT/M cells at all stages (85% of CT/M cells at E4, 72% at E6 and 66% at E10). The fraction of *PRRX1+* cells among CT/M cells was maintained at 66%, while the fraction of *PRRX1+* cells among all limb cells dropped with time from 88% at E4 to 25% at E10 (Supplementary Fig. 8c), suggesting that *PRRX1* expression is actively maintained in CT/M cells. This is consistent with the contribution of *Prrx1* lineage to mouse postnatal muscles[41] (Yaseen Badarneh et al., back-to-back submission). *TWIST2*

**Fig. 4 Identification of a cell population displaying a dual muscle/CT transcriptional identity in chicken limb cells at foetal stages. a** UMAP plots showing the whole-limb clustered populations with their respective number of cells at E4 (4820 cells), E6 (3340 cells) and E10 (22912 cells). **b** Heatmaps showing the relative expression of the top 10 markers for each CT and muscle clusters in cells grouped by their identity CT (red), CT/M (green) or M (blue) at E4, E6 and E10. Upregulated genes in yellow, downregulated genes in purple. CT/M identity labels 0.5% of limb cells at E4, 4.0% of limb cells at E6 and 3.3% of limb cells at E10. **c** Standard box plots showing the CT and M scores in cells grouped by their CT (red), M (blue) identity at E4, E6 and E10. The lower and upper hinges correspond to the first and third quartiles (the 25th and 75th percentiles). The upper whisker extends from the hinge to the largest value no further than 1.5 * IQR from the hinge (where IQR is the inter-quartile range). The lower whisker extends from the hinge to the smallest value at most 1.5 * IQR of the hinge. The statistical test used is the two-sided Wilcoxon rank sum-test. Sample size: $n = 4758$ cells from two E4 embryos, 3268 cells from three E6 embryos and 21256 cells from three E10 embryos. The exact $p$-values are indicated on the box plots. **d** Violin plots showing Log-normalised expression levels of representative CT markers (CDH11 at E6, COL3A1 at E10) and muscle markers (SLIT1 at E4, FGFR4 at E6, C1QTNF3 at E10) in cells grouped by their CT (red), CT/M (green) or M (blue) identity. The statistical test used is the two-sided Wilcoxon rank sum-test. Sample size, $n = 4758$ cells from two E4 embryos, 3268 cells from three E6 embryos and 21256 cells from three E10 embryos. The exact $p$-values are indicated on the violin plots. **e** Feature plots showing the distribution of cells with a CT/M identity (in blue) across whole-limb clustered populations (in grey) at E4, E6 and E10.

was also expressed in CT/M cells (15% at E4, 25% at E6 and 29.5% at E10), indicating that a TWIST2 cell population contributes to developmental muscles in addition to adult skeletal muscles as previously described[40]. The myogenic markers, *PAX7* and *MYOD1* showed similar distributions in CT/M cells, as a majority of CT/M cells expressed *PAX7* (42–57%) and *MYOD1* (36–57%), while much fewer cells expressed *MYOG* (10–15%). These observations are suggestive of an immature state of these CT/M cells (Supplementary Fig. 8c).

From this scRNAseq analysis, we conclude that limb cells exhibit a small population with a CT fibroblast/myogenic signature that is in transit between a CT phenotype and a muscle fate.

**BMP signalling promotes a fibroblast-to-myoblast conversion.** Cell communication signals, acting through secreted ligands, govern a multitude of cell-fate choices during development and postnatal life. To explore the mechanisms underlying the fibroblast-to-myoblast switch we examined which signalling pathways were enriched in the CT/M cells. *ID* genes are recognised transcriptional readouts of BMP activity[42]. *ID1*, *ID2* and *ID3* genes were expressed at a similar or greater level in CT/M cells compared to CT and M cells (Fig. 5a). In addition, the fraction of ID+ cells was significantly higher in CT/M cells versus CT and M cells at E6 and E10 (Fig. 5b). Consistently, *ID2* expression was enriched at the CT/muscle interface, labelled with *SCX* expression in E6 chicken limbs (Fig. 5c, d). The nuclear location of pSMAD1/5/9, another readout of active BMP, was increased at the periphery of dorsal and ventral muscle masses close to sites of *BMP4* expression (Fig. 5e–h) and at muscle tips (Fig. 5i) at E6. By the time the muscle pattern was fully set, i.e. between E7 and E9, *ID2* transcripts and pSMAD1/5/9 were observed at the tips of myotubes (myosins+) close to tendons on longitudinal or transverse sections (Fig. 5j–n). Interestingly, in our quail-chick grafting experiments, we detected 2 populations of pSMAD1/5/9+ myonuclei at muscle tips, one somite-derived (yellow arrows in Fig. 6) and one non-somite-derived (white arrows in Fig. 6) at both E6 and E9. This is reminiscent of the two distinct myonucleus populations recently identified in adult mouse muscles at the myotendinous junction, with one of them being enriched in CT-associated collagen genes[43]. Thus, our findings support the notion that myonuclei with a fibroblast signature could be of CT origin. Altogether, these results show the existence of a BMP-responsive population with a dual CT/muscle identity that could correspond to the fibroblast-derived myogenic cells found at the CT/muscle interface.

Based on BMP activity in this junctional cell population in chicken limbs (Fig. 5), we hypothesised that BMP signalling would be involved in the recruitment of a subpopulation of CT

fibroblasts towards a muscle fate. To test the ability of BMP signalling to control a fibroblast-to-myoblast conversion, we modified BMP activity in chicken limbs and assessed the consequences for PAX7+ myogenic progenitors and TCF4+ CT fibroblasts (Fig. 7a–i). Exposure of chicken limbs to retroviral-BMP4 led to an increase in the number of PAX7+ muscle progenitors (Fig. 7e, f) as previously observed[23]. However, consistent with the cell-fate conversion hypothesis, the increase of PAX7+ progenitors occurred at the expense of TCF4+ fibroblasts (Fig. 7i) with no change in the total cell number (Fig. 7g, h). Moreover, no change in the proliferation rate of PAX7+ cells was observed upon BMP4 exposure in chicken limbs (Supplementary Fig. 9a–g). This abnormal increase in PAX7+ cells led to muscle patterning defects, such as muscle splitting defects visualised by the fusion of individual muscles (Fig. 7c–f), which was probably due to the reduction of TCF4+ fibroblasts known to control muscle patterning[19]. BMP overexpression was also applied to myoblast primary cultures in proliferation culture conditions. Consistent with our conversion hypothesis, BMP4 treatment did not alter the number of PAX7+ cells in myoblast cultures (Supplementary Fig. 9h–j). To assess the fibroblast-to-myoblast conversion, we applied retroviral-expressing BMP4 or BMPR1Aca (a constitutive active form of the BMPR1A) to CEFs (chicken embryonic fibroblasts) that were isolated from whole embryos. BMP over-activation induced the appearance of PAX7+ cells compared to control cultures, albeit with a low efficacy (Fig. 7j–m). This shows that a subset of this mixed fibroblast population can convert into PAX7+ cells when exposed to BMP in vitro.

We then performed BMP loss-of-function experiments in chicken limbs by overexpression of the BMP antagonist NOGGIN. This approach induced the converse phenotype, i.e. an increase in the number of TCF4+ fibroblasts at the expense of PAX7+ muscle progenitors with no global change in cell number (Fig. 8a–i). The behaviour of TCF4+ fibroblasts upon BMP misexpression is consistent with the respective decrease and increase in the expression of the *OSR1* CT-marker upon BMP gain- and loss-of-function experiments in chicken limbs[44]. Similar to the in vivo experiments in chicken embryos, the inhibition of BMP signalling in myoblast cultures led to a fibroblast phenotype (Fig. 8j–n and Supplementary Fig. 9k, l). BMP inhibition was achieved with the overexpression of SMAD6 (Fig. 8j–n), an inhibitor of the pSMAD1/5/9 pathway[42] or by using the BMP antagonist NOGGIN (Supplementary Fig. 9k, l). BMP inhibition led to a reduction in the number of PAX7+, MYOD+ and MYOG+ myogenic cells and myotubes in myoblast cultures in differentiation culture conditions (Fig. 8j–l). This was associated with a decrease in the mRNA expression levels of muscle lineage markers from progenitor to differentiation state (*PAX7*, *MYF5*, *MYOD*,

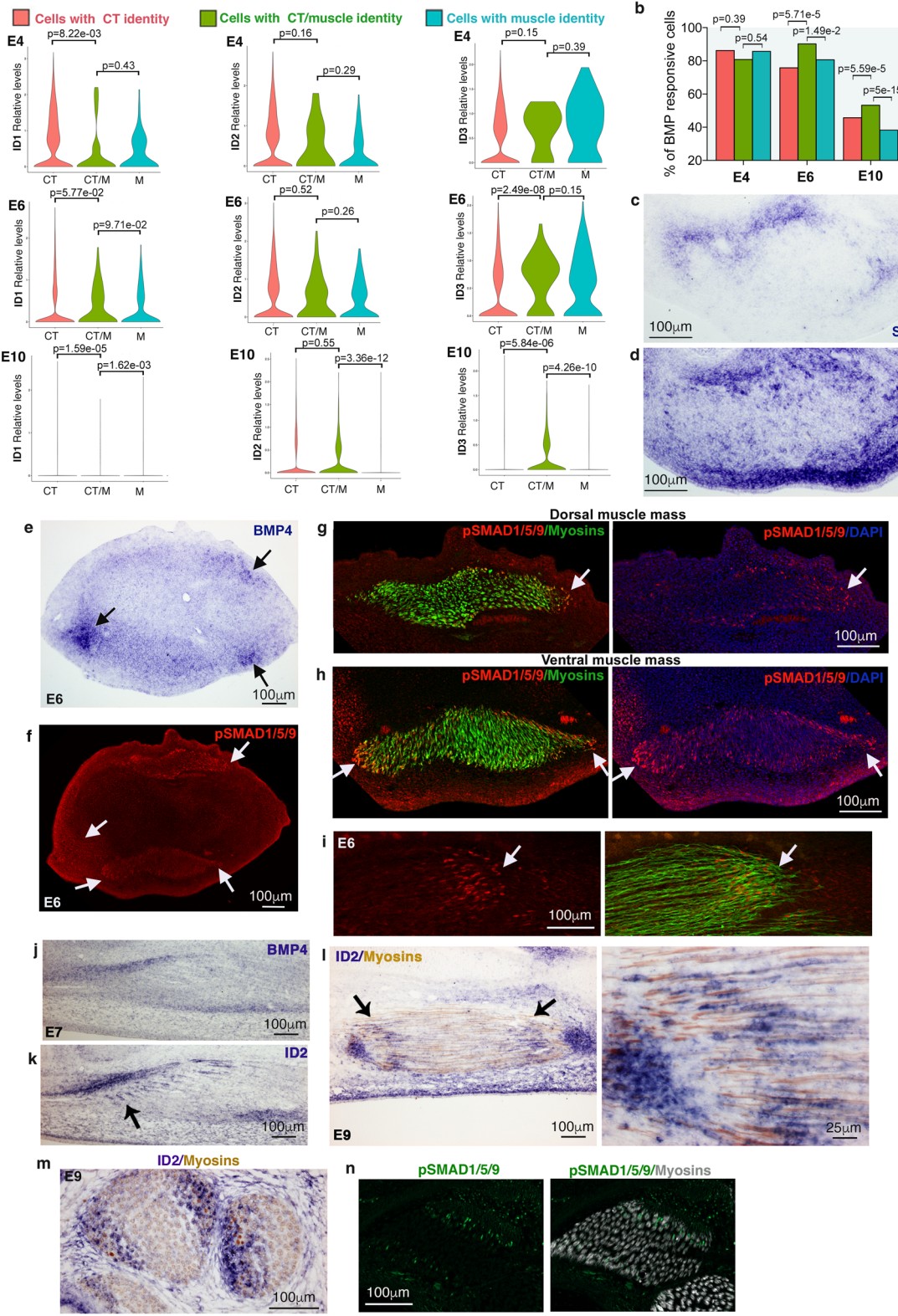

*MYOG, MYHC*) (Fig. 8m). The SMAD6-treated muscle cells adopted a fibroblast-like phenotype compared to myotubes observed in control myoblast cultures (Fig. 8j–l) and displayed an increased expression of CT markers, such as *COL3A1, COL6A1* and *OSR1* (Fig. 8n). These results show that BMP inhibition converts myoblasts towards a fibroblast phenotype.

Altogether, these in vivo and in vitro BMP functional experiments unravel an unexpected role for BMP signalling in driving a fibroblast/myoblast switch during limb muscle patterning.

## Discussion

Here, we identified a subpopulation of limb myogenic cells that did not originate from the somitic *Pax3*-lineage in chicken and mouse embryos. This subpopulation of myogenic cells originated from lateral plate-derived cells (chicken embryos) and CT-marker

**Fig. 5 BMP signalling is active in cells displaying a CT/M transcriptional identity and in cells at the muscle/tendon interface in chicken limbs. a** Violin plots showing Log-normalised expression levels of *ID1*, *ID2* and *ID3* genes in cells grouped by their CT (red), CT/M (green) or M (blue) identity at E4, E6 and E10. The statistical test used is a two-sided Wilcoxon rank sum-test; Sample size: $n = 4758$ cells from two E4 embryos, 3268 cells from three E6 embryos and 21256 cells from three E10 embryos. The exact *p*-values are indicated on the violin plots. **b** Bar plot showing the percentage of cells expressing *ID* genes in CT (red), CT/M (green) or M (blue) cells at E4, E6 and E10. The statistical test used is the two-sided Fisher's exact test. Sample size: $n = 4758$ cells from two E4 embryos, 3268 cells from three E6 embryos and 21256 cells from three E10 embryos. The exact *p*-values are indicated on the graphs. **c, d** Focus on ventral limb muscle masses from adjacent and transverse limb sections of E6 chicken embryos hybridised with SCX (**c**) and ID2 (**d**) probes. **e, f** Transverse adjacent limb sections of E6 chicken embryos were hybridised with BMP4 probe (**e**) and immunostained with pSMAD1/5/9 antibody (red) (**f**). **e, f** Arrows point to BMP4 and pSMAD1/5/9 expression spots. **g, h** High magnification of dorsal (**g**) and ventral (**h**) limb muscle masses immunostained with pSMAD1/5/9 (red) and MF20 (myosins, green) antibodies and counterstained with DAPI, arrows point to pSMAD1/5/9 expression spots at muscle periphery. **i** Longitudinal view of a muscle labelled with pSMAD1/5/9 (red) and MF20 (myosins, green) antibodies, arrows point to pSMAD1/5/9 enriched expression at muscle tips. **j, k** Adjacent and longitudinal limb sections of E7 chicken embryos hybridised with BMP4 (**j**) or ID2 (**k**) probes (blue staining), arrow points to *ID2* expression at muscle tips close to tendons (labelled with *BMP4* and *ID2* transcripts). **l** Longitudinal muscle sections of E9 chicken hybridised with ID2 probe (blue) and immunostained with MF20 antibody (myosins, brown). The arrow points to *ID2* expression at muscle tips. High magnification of one muscle tip. **m** Transverse limb sections of E9 chicken embryos were hybridised with ID2 probe (blue) and immunostained with MF20 (myosins, brown) antibody. Focus on dorsal limb muscles. **n** Transverse limb sections of E9 chicken embryos immunostained with pSMAD1/5/9 (green) and MF20 (myosins, grey) antibodies. Focus on dorsal limb muscles.

lineages (mouse embryos). Based on quantification in chicken and mouse developing limb muscles, this myogenic population of CT fibroblast origin represents 4–9% of muscle cells and was preferentially located at muscle tips close to tendons during foetal myogenesis. Interestingly, the recruitment of mesenchymal stromal cells has already been observed in the myotome in somites of mouse embryos[45], although the regionalisation of the recruitment had not been addressed in that study. The unexpected contribution of lateral plate-derived CT fibroblasts to the muscle lineage opens ~~new~~ perspectives for the mechanisms underlying limb muscle patterning. Previous BrdU/EdU-based experiments have shown preferential incorporation of nuclei in myotubes at muscle tips during foetal and postnatal limb myogenesis[46–48]. Although these BrdU/EdU-based experiments never addressed the origin of the newly incorporated nuclei, it was always assumed that this preferential incorporation at the muscle tips was from somite-derived cells. We now propose that the incorporation of new nuclei at muscle tips of forming myotubes also includes CT lineage-derived nuclei. As the lateral plate-derived cells are recognised to drive limb muscle patterning[1–7], we see these muscle cells of CT origin as a source of positional information for limb muscle patterning. One compelling hypothesis is that these CT lineage-derived muscle cells would regulate the location of myoblast fusion events and thereby direct the shape and orientation of limb muscle growth during foetal development. These CT lineage-derived muscle cells are reminiscent of muscle founder cells that drive muscle identity and position in *Drosophila* embryos[49]. In addition to playing a role in limb muscle patterning, these regionalised myonuclei of CT origin could participate to myotendinous junction formation. Consistent with this idea, the myonuclei at the myotendinous junction display a fibroblast gene signature[43]. A possible scenario would be that the CT fibroblasts incorporated in the myotubes at muscle tips close to tendons maintain a fibroblast genetic programme in addition to being reprogrammed to muscle. This molecular fibroblast signature enriched in extracellular matrix components[43] could then participate in myotendinous junction formation.

The reprogramming of non-muscle nuclei to muscle nuclei after fusion has been already demonstrated in heterokaryons in vitro[50]. Moreover, reprogramming of mesenchymal stem cell nuclei to muscle nuclei was shown to rely on fusion and to be mediated by IL4/NFAT signalling[45]. Based on our data, we envision an additional step of fibroblast-to-muscle conversion before fibroblast incorporation to myotubes during limb foetal myogenesis. Our data supporting this scenario are (1) the

identification of PAX7+ muscle progenitors that were of *Scx* or *Osr1* lineage in mouse limbs, (2) the identification of a sub-population of chicken limb cells with a dual CT/muscle identity by scRNAseq analysis, (3) and the ability of BMP signalling to tilt the balance between muscle and CT fate. Interestingly, this limb cell population with a dual CT/muscle identity is maintained in postnatal mouse limb muscles[41] (Yaseen Badarneh et al., back-to-back submission). These dual CT/muscle identity cells can be related to the bi-fated cells identified at the bone/tendon insertion in mice during development[51].

We identified BMP as molecular signalling involved in the recruitment of CT fibroblast towards the myogenic lineage in chicken limb cells. BMP signalling is active at the right place at the CT/muscle interface during foetal development. The presence of BMP activity in this junctional cell population combined with the BMP gain- and loss-of-function experiments in chicken limbs and in cell cultures lead us to propose a scenario in which BMP locally drives the conversion of CT fibroblasts to myoblasts, which are incorporated into myotubes at muscle tips close to tendons. Persistent BMP signalling regionalised at muscle tips could provide an in vivo safeguard to maintain the muscle identity of fibroblast-derived muscle cells. We believe that the BMP effect on fibroblast-muscle conversion during foetal myogenesis is unrelated to the BMP function in the equilibrium between PAX7+ progenitor expansion and differentiation during postnatal growth[52–54]. However, the BMP effect on fibroblast/myoblast conversion during foetal myogenesis could be related to the function of BMP signalling in the specification of PAX7+ cells in the dorsal lateral plate tissue (lateral plate equivalent) and not in the paraxial mesoderm (somite equivalent) in *Xenopus* embryos[55], although we are not looking at the same stage or process. Consistently, BMP signalling is also enriched in the dual CT/muscle cells in postnatal mouse limb muscles[41] (Yaseen Badarneh et al., back-to-back submission). Interestingly, BMP signalling has been already shown to be involved in the formation of another junctional population at the level of the tendon/bone insertion, by regulating *Scx* expression during mouse limb development[56]. Indeed, BMP signalling has been described in numerous processes related to cell-fate control and reprogramming during development, wound healing and pathologies[57–61].

In summary, our studies show that the generation of the tendon/muscle interface partly results from a fibroblast-to-myoblast conversion driven by localised BMP signalling during foetal myogenesis. This unexpected scenario provides a ~~novel~~ cellular and molecular mechanism underlying limb muscle patterning and myotendinous junction formation. The existence of dual

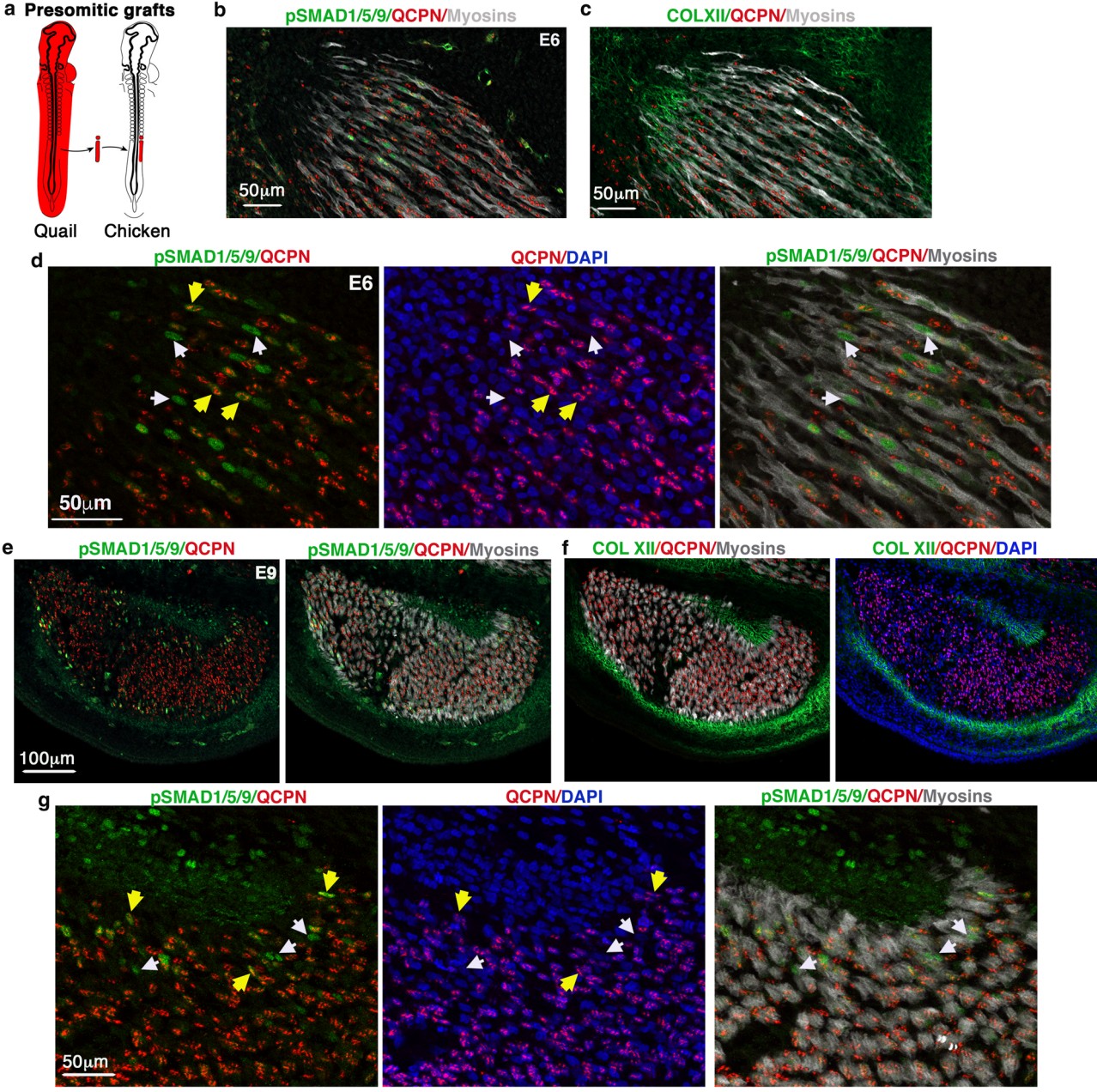

**Fig. 6 PSMAD1/5/9+ myonuclei label two myonucleus populations of distinct embryological origins at muscle tips. a** Schematic of isotopic presomitic mesoderm grafts from quail to chicken embryos. **b** Longitudinal view of a limb muscle from E6 chimera labelled with pSMAD1/5/9 (green), QCPN (quail nuclei, red) and MF20 (myosins, grey) antibodies, with a focus on muscle tips. **c** Adjacent section to **b** labelled with COLXII (tendon, green), QCPN (quail nuclei, red) and MF20 (myosins, grey) antibodies. **d** Focus on muscle tips, labelled with pSMAD1/5/9 (green), QCPN (quail nuclei, red) and MF20 (myosins, grey) antibodies, combined with DAPI (blue). White arrows point to pSMAD1/5/9+ myonuclei (green) that are not of quail origin (QCPN−), while yellow arrows point to pSMAD1/5/9+ myonuclei (green) that are of quail origin (QCPN+, red). **e** Transverse view of the FCU muscle from E9 chimera labelled with pSMAD1/5/9 (green), QCPN (quail nuclei, red) and MF20 (myosins, grey) antibodies. **f** Adjacent section to **e** labelled with COLXII (tendon, green), QCPN (quail nuclei, red) and MF20 (myosins, grey) antibodies, combined with DAPI. **g** Focus on muscle tips. White arrows point to pSMAD1/5/9+ myonuclei (green) that are not of quail origin (QCPN−), while yellow arrows point to pSMAD1/5/9+ myonuclei (green) that are of quail origin (QCPN+, red).

identity cells could be part of a universal mechanism to generate tissue junctions and deal with cell heterogeneity to finally shape developing adjacent tissues. However, it is not yet clear whether the existence of dual identity cells would be part of the original mechanism underlying muscle patterning or whether it is a later evolutionary acquisition to enable the remarkable variation in muscle shapes and size that occurred through mammalian evolution. Future studies will be also needed to explore whether this mechanism can be reactivated in adult tissues and where it could be relevant in the context of muscle regeneration.

## Methods

**Chicken and quail embryos**. Fertilised chicken and quail eggs from commercial sources (White Leghorn and JA57 strain, Japanese quail strain, EARL Les Bruyères, Dangers, France) were incubated at 37.5 °C. Before 2 days of incubation, chicken and quail embryos were staged according to somite number. After 2 days of

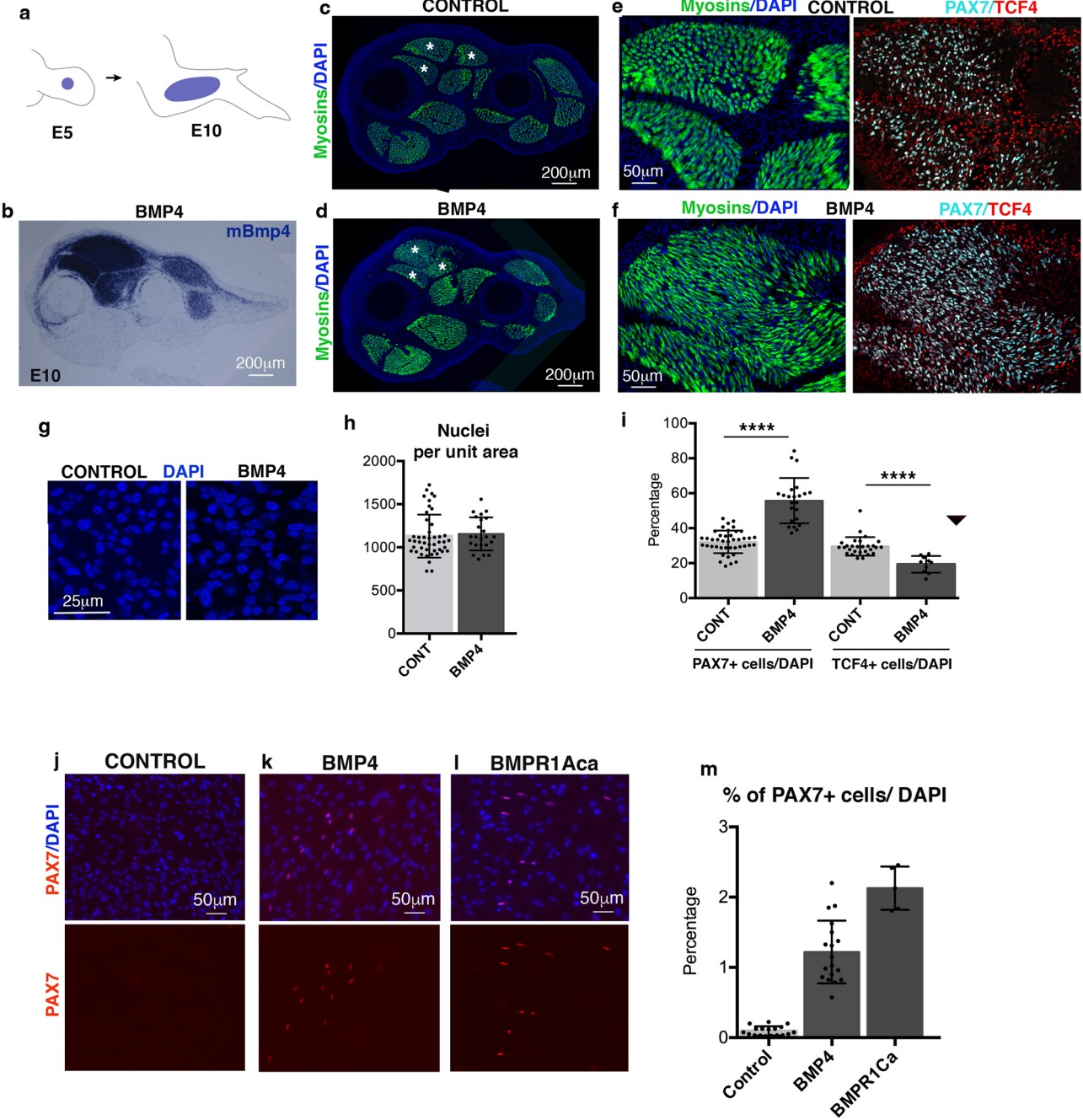

**Fig. 7 BMP gain-of-function experiments in limbs and fibroblast cultures. a–i** BMP4/RCAS-producing cells were grafted to the right limbs of E5 chicken embryos (**a**). Embryos were fixed 5 days later at E10. In situ hybridisation experiments with mBmp4 probe to transverse limb sections indicate the extent of ectopic BMP4 expression in dorsal limb muscles (**b**). **c–f** Limb sections were immunostained with MF20 (myosins, green) to visualise the muscle pattern. Asterisks indicate fused muscles in BMP4-grafted limbs (**d**) versus individual muscles in control limbs (**c**). **e, f** High magnifications of dorsal limb muscles immunostained with MF20 (myosins, green), PAX7 (cyan) and TCF4 (red) antibodies in BMP4 (**f**) and control limbs (**e**). **g, h** Representative fields of nuclei (DAPI+) in muscles of control and BMP4 limbs (**g**) and quantification of the number of nuclei per unit area (**h**). **h** Data are presented as mean of values ± SD from 5 BMP4 and 9 control limbs. **i** Percentages of PAX7+ and TCF4+ nuclei per total number of nuclei in BMP4-grafted limbs versus controls. Data are presented as mean values ± SD from 3 BMP4 and 6 control limbs. The p-value was obtained using the two-tailed Mann–Whitney test. Asterisks indicate the p-values, ****p < 0.0001. **j–l** Chicken fibroblast cultures transfected with empty/RCAS (control) (**j**), BMP4/RCAS (**k**) or BMPR1ca/RCAS (**l**). Immunostaining with PAX7 antibody (red) and DAPI in BMP4- and BMPR1ca-transfected fibroblast cultures compared to controls. **m** Quantification of PAX7+ cells versus the total cell number. Data are presented as mean values ± SD from 3 independent experiments.

incubation, embryos were staged according to days of in ovo development. GFP + chicken were generated and provided by the Roslin Institute[26].

**Mouse strains**. Animals were handled according to European Community guidelines, implementing the 3Rs rules. Protocols were validated by the ethic committee of the French Ministry, under the reference numbers APAFIS#13695-

2018021408521124.v2, APAFIS#24357-2020041613396163.v3 and APAFIS#6354-20160809l2028839.v4 and Institut Pasteur ethics committee (CETEA, reference 2015–0008). The following strains were described previously: *Pax3^{Cre 26}*, *Wnt1^{Cre 37}*, *Scx^{Cre 56}*, *Osr1^{GCE}* allele (*Osr1^{eGFP-CreERT2}*)[15], *Scx^{GFP 62}*, *Rosa26^{Tomato}*(Ai9)[28,29], *Rosa26^{LoxP-stop-LoxHTB 30}*, and Fucci2AR[35] (Supplementary Table 3). For temporal fate mapping using the *Osr1^{GCE}* allele, we used the lineage-specific *Pax7*-driven

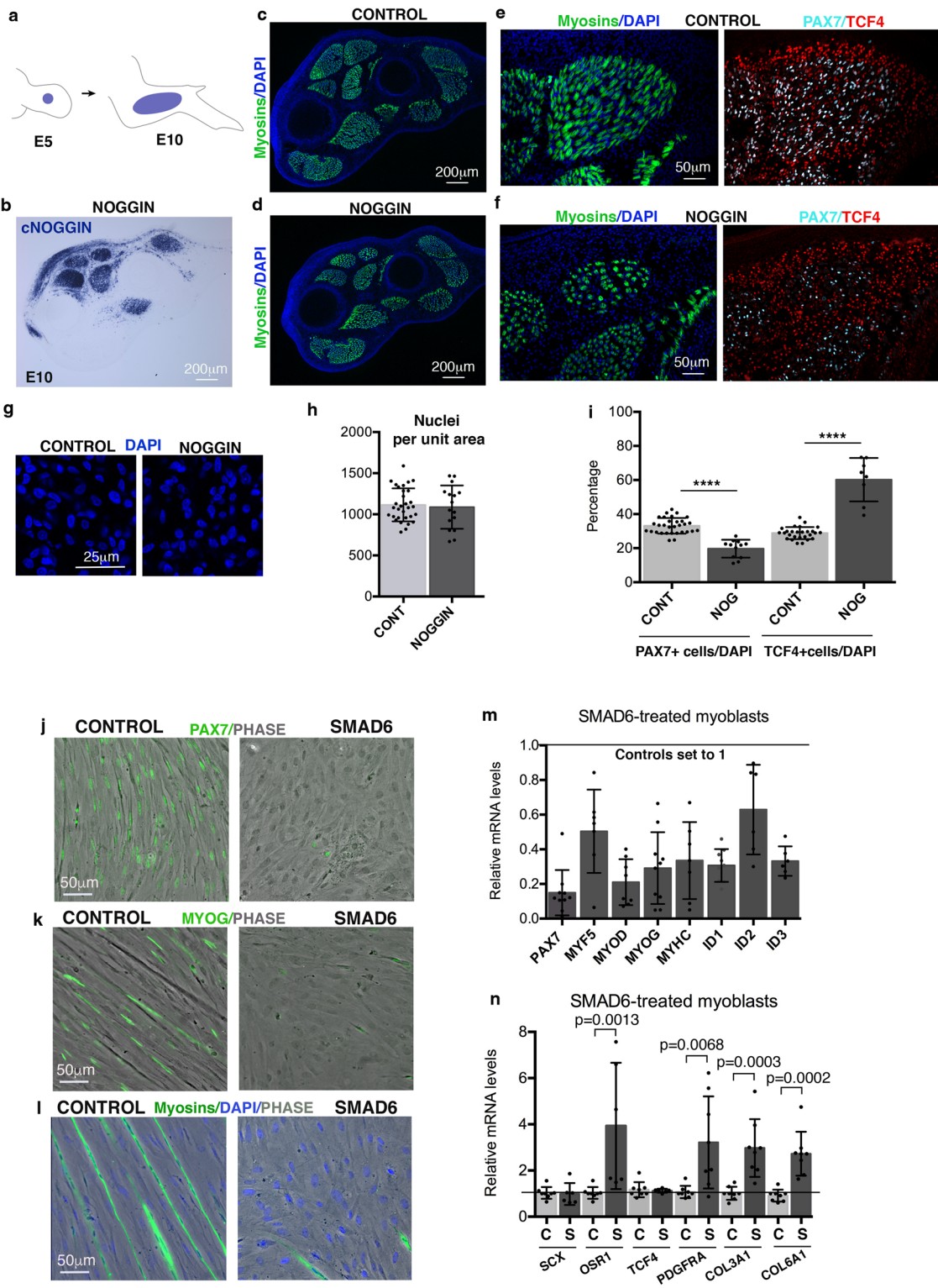

reporter (*Pax7^{nGFP-stop-nlacZ}*) named *Pax7^{GPL}*, in which locus accessibility to Cre-mediated recombination allows higher sensitivity for tracking Pax7-expressing cells and their progeny[34]. Adult mice were kept in ventilated cages on a 12-h/12-h light/dark cycle, 22 °C ambient temperature and 34% humidity. Male and female adult mice (*Mus musculus*, 6–12 weeks of age) were kept on mixed genetic backgrounds C57BL/6JRj or DBA/2JRj (B6D2F1, Janvier Labs). To induce recombination, 5 mg of tamoxifen (Sigma #T5648) were administered by gavage to pregnant females at different time points between E10.5 and E13.5. A 25 mg/ml stock solution in 5% ethanol and 95% sunflower seed oil was prepared by thorough resuspension with rocking at 4 °C.

**Surgical grafting experiments in avian embryos**

*Isotopic/Isochronic quail- or GFP+ chicken-into-chicken grafts of presomitic meso-derm.* Quail, GFP+ chicken and chicken embryos were incubated until they reached 15-somite stage. Of note, limb buds are not formed at this stage and brachial somitic cells have not yet migrated into the forelimb presumptive regions. An artificial dark field was obtained by injecting Indian ink, diluted 1:1 in PBS, beneath the chicken host embryos. Microsurgery was performed on the right side of the host embryos. The 15th somite and the non-segmented presomitic meso-derm was removed over a length corresponding to 5–8 somites. Presomitic mesoderms from either quail (*N* = 6) or GFP+ chicken (*N* = 4) donor embryos were rinsed in DMEM (PAA Laboratories)/10% foetal calf serum (FCS, Eurobio) and transplanted into chicken hosts submitted to the same ablation. Grafts were

**Fig. 8 BMP loss-of-function experiments in limbs and myoblast cultures. a–f** NOGGIN/RCAS-producing cells were grafted to right limbs of E5 chicken embryos (**a**). Embryos were fixed 5 days later at E10. **b** In situ hybridisation experiments with cNOGGIN probes to transverse limb sections indicate the extent of ectopic NOGGIN expression in dorsal limb muscles. **c–f** Limb sections were immunostained with MF20 (myosins, green) to visualise muscles. **e, f** High magnifications of dorsal limb muscles immunostained with PAX7 (cyan) and TCF4 (red) antibodies in NOGGIN and control limbs. **g, h** Representative fields of nuclei (DAPI+) in muscles of control and NOGGIN limbs (**g**), and quantification of the number of nuclei per unit area (**h**). **h** Data are presented as mean of values ± SD from 4 NOGGIN and 4 control limbs. **i** Percentages of PAX7+ and TCF4+ nuclei per total number of nuclei in NOGGIN-grafted limbs versus controls. Data are presented as mean values ± SD from 3 NOGGIN and 6 control limbs. The *p*-value was obtained using the two-tailed Mann–Whitney test. Asterisks indicate the *p*-values, ****$p < 0.0001$. **j–n** Myoblast cultures transfected with empty/RCAS (control) or SMAD6/RCAS were induced to differentiation. **j–l** Immunostaining with PAX7 (**j**), MYOG (**k**), and MF20 (myosins) (**l**) antibodies (green) in control and SMAD6 cultures. **m, n** RT-qPCR analyses of the expression levels of muscle genes, *PAX7, MYF5, MYOD, MYOG, MYHC*, BMP-responsive genes *ID1, ID2, ID3* (**m**) and connective tissue genes, *SCX, COL3A1, COL6A1, OSR1, TCF4, PDGFRA* (**n**) in SMAD6-transfected and control cell cultures. Gene mRNA levels were normalised to *GAPDH* and *RPS17*. The relative mRNA levels were calculated using the $2^{-\Delta\Delta Ct}$ method using the control condition as controls. For each gene, the mRNA levels of the control condition were normalised to 1. Graphs show means ± standard deviations of 8 biological samples (**m**) and 6 biological samples (**n**). **n** The exact *p*-values were obtained using the two-tailed Mann–Whitney test and are noted on the graphs.

performed according to the original dorso-ventral and antero–posterior orientations (Fig. 1a–g and Supplementary Figs. 1, 2, 9).

*Isotopic/isochronic quail-into-chicken grafts of limb lateral plate.* Quail and chicken embryos were incubated until they reached 18-somite stage. Of note, limb buds are not formed at this stage, and brachial somitic cells have not yet migrated into the forelimb lateral plate. The forelimb lateral plate mesoderm at the level of the 15th somite and over a length of 3 somites was excised from host chicken embryos and replaced by their quail counterpart ($N = 3$) (Fig. 1h–k and Supplementary Fig. 3).

*Isotopic/isochronic GFP + chicken-into-chicken grafts of neural tube.* GFP + chicken and chicken embryos were incubated until they reached 12- to 14-somite stage. At this stage, neural crest cells did not yet leave the neural tube to colonise the limb bub. The neural tube was excised over a length corresponding to 5–8 somites from GFP + chicken donor embryos and grafted in place of the neural tube of chicken host embryos ($N = 4$) (Supplementary Fig. 6).

For all types of grafts, quail-chicken and GFP + chicken–chicken chimeras were allowed to grow for another 4–8 days and treated for either immunohistochemistry or in situ hybridisation to tissue sections.

**Lateral plate electroporation.** Chicken embryos were incubated until they reached 19/20-somite stage (E2). The DNA solution was composed of both pT2AL-CMV/βactin-membrane/Tomato-T2A-H2B/GFP and CMV/βactin-Transposase at a molar ratio of 1:3. The transposase allows the stable integration of membrane-tomato and nuclear-GFP into the chicken genome[27]. The DNA solution was electroporated with 3 pulses of 50 V, 5 ms duration, 50 ms interpulse interval, between two electrodes of 0.3 mm (negative) and 0.5 mm (positive) diameter tungsten rods. Pulses were delivered with a TSS20 electroporator and a EP21 current amplifier (Intracell). Embryos were fixed at E9, 7 days after electroporation.

**Grafting experiments in chicken limbs.** Chicken embryonic fibroblasts (CEFs) obtained from E10 chicken embryos were transfected with BMP4-RCAS[23] or NOGGIN-RCAS[23] using the Calcium Phosphate Transfection Kit (Invitrogen, France). Cell pellets of ~50–100 μm in diameter were grafted into the limbs of E5 embryos. BMP4-RCAS- ($N = 8$) and NOGGIN-RCAS- ($N = 4$) grafted embryos were harvested at E9/E10 and processed for in situ hybridisation and immunohistochemistry.

**EdU labelling.** Fifty microliters of EdU (Invitrogen, A10044) solution (5 mg/ml) was injected into the circulation in E10 chick embryos for 1.5 h. Embryos were then fixed and immunohistochemistry was performed on limb transverse sections. EdU was detected using the Click-iT Plus EdU Cell Proliferation Kit for Imaging (Invitrogen, C10639).

**Immunohistochemistry.** For antibody staining, control and manipulated chicken forelimbs were fixed in a paraformaldehyde (PFA) 4% solution in PBS overnight, embedded in gelatin/sucrose and then cut in 12-μm cryostat sections. Mouse tissues were fixed for 2 h in 4% PFA 0.1% triton at 4 °C, washes extensively in PBS, equilibrated in 30% sucrose in PBS, and embedded in OCT for cryo-sectioning. All antibodies (sources, conditions of use, references) are reported in Supplementary Table 3. Quail cells were detected using the QCPN monoclonal antibody (DSHB cat. # QCPN, undiluted supernatant). Differentiated muscle cells were detected using the MF20 monoclonal antibody, MF20 (DSHB cat. # MF20, undiluted supernatant), recognising sarcomeric myosin heavy chains. Mouse myoblasts were detected using the MYOD monoclonal antibody (BD Biosciences cat. # 554130, 1:500) or with a mixture of MYOD and MYOG monoclonal antibodies (DSHB cat. # F5D, 1:30). Chicken myoblasts were detected using the MYOG polyclonal antibody[63] (undiluted). Muscle progenitors were detected using the PAX7

monoclonal antibody (DSHB cat. # AB_PAX7, 1:200). Active BMP signalling was detected using the pSMAD polyclonal antibody (Cell signalling cat. # 9516, 1:100) recognising the complex BMP-activated receptor-phosphorylated SMAD1/5/9. Tendons and connective tissues, were detected using Collagen type XII polyclonal antibody[64] (Clone #522, 1:100) and TCF4 polyclonal antibody (Cell signalling cat. # 2569, 1:100), respectively. Cells from lineage tracing experiments were labelled with the anti-beta-gal (MP Biomedicals cat. # MP 55976, 1:1500), anti-GFP (Abcam cat. # ab13970, 1:1000) and anti-Tomato (Takara cat. # 632496, 1:400) polyclonal antibodies. The following secondary antibodies were used: goat anti-mouse IgG coupled to Alexa Fluor 555 (Invitrogen cat. # A21422, 1:200), goat anti-mouse IgG1 coupled to Alexa Fluor 488 (Jackson ImmunoResearch Labs cat. # 155-545-205, 1:500), Alexa Fluor 633 (Thermo Fisher Scientific cat. # A-21126, 1:500), Alexa Fluor 647 (ThermoFisher Scientific cat. # Z-25008, 1:500) or Cy3 (Jackson ImmunoResearch Labs cat. # 115-165-205, 1:500), goat anti-mouse IgG2 coupled to Alexa Fluor 488 (Invitrogen cat. # A21141, 1:200), Alexa Fluor 633 (ThermoFisher Scientific cat. # A-21146, 1:500) or Alexa Fluor 647 (ThermoFisher Scientific cat. # A-21242, 1:500), goat anti-rabbit IgG coupled with Alexa Fluor 488 (ThermoFisher Scientific cat. # A-11070, 1:500), Alexa Fluor 555 (Thermo Fisher Scientific cat. # A-21430, 1:500) or Alexa Fluor 633 (Thermo Fisher Scientific cat. # A-21072, 1:500) and goat anti-chicken IgG coupled to Alexa Fluor 488 (Thermo Fisher Scientific cat. # A-11039, 1:500). To label nuclei, sections were incubated for 15 min with DAPI (SIGMA-Aldrich cat. # D9542, 1:1000). Stained sections were examined using ApoTome.2 (Zeiss), SP5 AOBS confocal (Leica) or LSM700 confocal (Zeiss) microscopes.

**In situ hybridisation to tissue sections.** Normal or manipulated embryos were fixed in 4% PFA. Limbs were cut in 12-μm cryostat transverse sections and processed for in situ hybridisation. Alternating serial sections from embryos were hybridised with probe 1 and probe 2. The digoxigenin-labelled mRNA probes were used as described[23,44]: SCX, ID2, BMP4, NOGGIN (chicken probes), and mBmp4 (mouse probe).

**Single-cell RNA-sequencing analysis from limb cells.** For scRNAseq analysis of chicken limb cells at different developmental stages, forelimbs from 2 different E4 embryos, 3 different E6 embryos and 3 different E10 embryos were dissected and dissociated by collagenase and mechanical treatments. Cell concentration was adjusted to 5000 cells/μl in the buffer. 5000 cells per conditions were loaded into the 10× Chromium Chip with the Single Cell 3′ Reagent Kit v3 according to the manufacturer's protocol. Samples were pooled together at each developmental stage (E4 ($N = 2$), E6 ($N = 3$) and E10 ($N = 3$)) before making the libraries. Libraries were then sequenced by pair with a HighOutput flowcel using an Illumina Nextseq 500 with the following mode (150 HO): 28 base-pairs (bp) (Read1), 125 bp (Read 2) and 8 bp (i7 Index). A minimum of 50 000 reads per cell were sequenced and analysed with Cell Ranger Single Cell Software Suite 3.0.2 by 10× Genomics. Raw base call files from the Nextseq 500 were demultiplexed with the cellranger mkfastq pipeline into library-specific FASTQ files. The FASTQ files for each library were then processed independently with the cellranger count pipeline. This pipeline used STAR21 to align cDNA reads to the *Gallus gallus* genome (Sequence: GRCg6a, *Gallus gallus* reference). The used sample size in scRNAseq allowed us to robustly identify cell populations as small as 18 cells at E4, 6 cells at E6 and 66 cells at E10.

The Seurat package (v3.0)[65] under R (v3.6.1)[66] was used to perform downstream clustering analysis on scRNAseq data[67]. Cells went through a classical Quality Control using the number of detected genes per cell (nFeatures), the number of mRNA molecules per cell (nCounts) and the percentage of expression of mitochondrial genes (pMito) as cut-offs. Outliers on a nFeature vs nCount plot were manually identified and removed from the dataset. Most importantly for this study, potential doublets were identified by running the Scrublet algorithm[68] and then removed from the dataset. Gene counts for cells that passed the above

selections were normalised to the total expression and Log-transformed with the NormalizeData function of Seurat using the nCount median as a scale factor. Highly variable genes were detected with the FindVariableFeatures function (default parameters). The cell cycle effect was regressed out using the ScaleData function. Using highly variable genes as input, principal component analysis was performed on the scaled data in order to reduce dimensionality. Statistically significant principal components were determined by using the JackStrawPlot and the ElbowPlot functions. Cell clusters were generated with the FindNeighbors/FindClusters functions (default parameters except for the number of selected PCs). Different clustering results were generated at different resolutions and for different sets of PCs. Non-linear dimensional reduction (UMAP) and clustering trees using Clustree[69] were used to visualise clustering results and select the most robust and relevant result. The CT and muscle clusters represented the majority of limb cells (90–95% of limb cells), in addition to other clusters encompassing the expected cell populations present in developing limb tissues such as vessels/blood, innervation and ectoderm (Fig. 4a). Muscle clusters gather mononucleated muscle cells, since the plurinucleated myotubes are excluded from the single-cell isolation protocol. Markers for each cluster were found using the FindAllMarkers function of Seurat (using highly variable genes as an input, default parameters otherwise) that ran Wilcoxon rank-sum tests (p-val adjusted < 0.05). CT and M scores were calculated using the AddModuleScore function for the CT and muscle clusters, respectively.

For the analysis of the CT/M cells, cells were grouped according to 4 identities (CT, CT/M, M and Other). The CT/M identity is defined by the co-expression (i.e. gene Log-normalised count > 0) within a cell of at least one of the CT markers (PRRX1, TWIST2, PDGFRA, OSR1, SCX) with at least one of muscle markers (PAX7, MYF5, MYOD1, MYOG). CT identity is conferred to all cells allocated to a CT cluster, except the CT/M cells. M identity is conferred to all cells allocated to a muscle cluster, except the CT/M cells. « Other » identity is conferred to all cells to clusters that are neither CT nor muscle, except for the CT/M cells. Differentially expressed genes between the CT/M cells and the combined CT and M cells were found using the FindMarkers function of Seurat (using highly variable genes as an input, default parameters otherwise) that ran Wilcoxon rank-sum tests (p-val adjusted < 0.05). The scRNAseq datasets were then analysed with the angle of these 4 identities using Seurat tools such as Feature plots, Heatmaps and Violin plots.

**Cell cultures.** Chicken embryonic fibroblast (CEF) cultures were obtained from E10 chicken embryos. Embryos without heads and viscera were mechanically dissociated. After centrifugation, cells were plated. Myoblast primary cultures were obtained from limbs of E10 chicken embryos[36,70].

Empty/RCAS, BMP4/RCAS and BMPR1ACa/RCAS constructions[23] were transfected to primary fibroblast cultures. Transfected fibroblasts were left for 5–7 days in culture with cell splitting to allow virus spread. Empty/RCAS, BMP4/RCAS, BMPR1ACa/RCAS fibroblasts were then fixed and processed for immunohistochemistry.

Empty/RCAS, BMP4/RCAS, NOGGIN/RCAS and SMAD6/RCAS constructions[23] and FUCCI/RCAS[36] were transfected to myoblast primary cultures. Transfected myoblasts were left for 5–7 days in culture with cell splitting to allow virus spread. Transfected myoblasts were either analysed in proliferation or in differentiation culture conditions. Empty/RCAS, NOGGIN/RCAS, SMAD6/RCAS or FUCCI/RCAS myoblasts were fixed and processed for immunohistochemistry or RT-q-PCR assays to analyse gene expression.

**Quantitative real-time PCR.** qPCR was performed as described in[24]. Total RNAs were extracted from SMAD6 or control myoblasts. 500 ng of RNA was reverse-transcribed. qPCR was performed on the Applied Biosystems StepOnePlus Real-Time PCR System using SYBR Green PCR master Mix (Life Technology, cat. # 4385614) with primers listed in Supplementary Table 4. The relative mRNA levels were calculated using the $2^{-\Delta\Delta Ct}$ method[71]. The $\Delta Ct$ were obtained from Ct normalised with GAPDGH and RPS17 levels in each sample.

**Cell counting.** Cell counting was performed with Fiji (v2.1.0) software[72]. Quantifications were performed on at least 3–5 pictures per experiments taken randomly within each sample. Pictures encompassed the central zone and tips of the muscles. Four to six limbs were counted depending on the experiment. Although labelling was preferentially restricted to muscle tips, we counted all the muscle cells present on the picture to compare with the numbers derived from the scRNAseq analysis and because it was not possible to define the border between the central zone and the tips in a reproducible manner. For most experimental settings, the percentage of non-somite-derived/CT-derived XX+ myogenic cells were calculated versus the total number of XX+ myogenic cells (for example, *Scx*-derived PAX7+ cells versus total PAX7+ cells).

**Statistics and reproducibility.** GraphPad Prism 6 software was used for statistical analysis. Non-parametric tests were used to determine statistical significance, which was set at p-values < 0.05. Unless stated in the figure legend, immunohistochemistry experiments have been performed on at least three limbs.

**Reporting summary**. Further information on research design is available in the Nature Research Reporting Summary linked to this article.

## Data availability

All data are available within the Article and Supplementary Files. The scRNAseq datasets have been deposited in the NCBI Gene Expression Omnibus database (https://www.ncbi.nlm.nih.gov/geo/) with the accession code number GSE166981. The scRNAseq analysis was performed with standard pipelines in Seurat R packages, as described in the Methods. Scripts will be made available upon request. The data and detailed protocols in this manuscript are available from the authors upon request. Source data are provided with this paper.

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

## Acknowledgements

We thank Sophie Gournet for the illustration. We thank the Roslin Institute (Prof Helen Sang and Dr. Adrian Sherman) for providing us with GFP⁺ chicken eggs. The production of the GFP + chicken embryos was supported by grants from BBSRC and the Wellcome Trust. Sequencing and genome mapping of the scRNAseq datasets were performed by the GENOM'IC platform at the Cochin Institute (Paris). This work was supported by the CNRS, Inserm, SU, AFM and FRM. ARTbio was supported by the CNRS, SU, the Institut Français de Bioinformatique (IFB) and by a grant from the SIRIC CURAMUS.

## Author contributions

J.E.d.L. initiated the project, conducted the BMP experiments, contributed and analysed the mouse work and performed lateral plate electroporations. C.B. conducted lateral plate grafting experiments, lateral plate electroporations, in situ hybridisation and immuno-histochemistry to experimental chicken and mouse limbs. M.-A.B. contributed to somite grafting experiments, performed cell preparation for scRNAseq and analysed the mouse work. E.H. performed confocal imaging of chicken embryos and conducted the bioin-formatic analysis of the scRNAseq datasets. G.C. conducted the mouse work and immunohistochemistry for myogenic markers. L.Y. performed G.F.P. chicken (somite and neural tube) and quail (somite) grafting experiments. M.-C.D. performed lateral plate electroporation in chicken embryos. L.B. and S.M. contributed to the design of the scRNAseq analysis methods. R.S. provided the Scx:Cre embryos. S.N. and C.F.-T. performed qPCR on fibroblasts. TJ performed cell preparation for scRNAseq. J.E.d.L., E.H., G.C., L.Y., C.R., C.F.-T., S.T. and F.R. discussed the project and contributed to writing the manuscript. D.D. supervised the study, analysed the results, pictured the data and organised the figures, wrote the manuscript and acquired funding.

## Competing interests

The authors declare no competing interests.
