## [Peer Review File · Nature Communications]

Editorial Note: Parts of this peer review file have been redacted as indicated to remove third-party material where no permission to publish could be obtained.

REVIEWER COMMENTS

Reviewer #1 (Remarks to the Author):

In this paper, the authors use both chick/quail grafting and mouse lineage tracing experiments to revisit the origin of limb muscle and connective tissue. The authors identify a novel contribution of the lateral plate to the forming musculature, mostly to the part that connects with the connective tissues. This finding was further supported by a single-cell analysis, where a population of cells that express a combination of myogenic and connective tissue genes was found. Finally, the authors suggest that BMP signaling regulates this conversion of fibroblasts to myoblasts.

In contrast to the part that describes the contribution of lateral plate mesoderm to the developing muscle in the CT/muscle interface, the part on the role of BMP regulation is not clear. Overexpressed BMP activity in chicken limbs increased in the number of PAX7+ muscle cells and reduced TCF4+ cells. The authors state that "the increase of PAX7+ progenitors occurred at the expense of TCF4+ fibroblasts". However, unless I missed something, they provide no direct evidence that TCF4+ cells convert to PAX7+ cells. Inducing BMP overexpression in chick/quail embryos with grafted lateral plate could provide such evidence. Moreover, following this logic, the number of CT cells that can potentially differentiate into muscle cells is not restricted, implying that this phenomenon is not unique to a population that forms the CT/muscle interface. I think that the authors should better explain their argument in the Discussion. Is this a unique population, or can any fibroblast transform into myoblast? What restricts this transformation in vivo? How is the BMP pathway involved in this activation/repression? These questions should be addressed. Finally, BMP4 was previously shown to regulate the differentiation of cells that connect between tendons and bones. It is important to discuss these findings and compare them to those of the current study. Overall, following the requested revisions, I support the publication of this manuscript.

Reviewer #2 (Remarks to the Author):

In this paper, Esteves de Lima et al. examine the possibility that fibroblasts originating from the lateral plate mesoderm undergo a fibroblast to muscle conversion, and become integrated into growing myofibres during embryonic development. Their experiments indicate that this contribution is under the control of BMP signals.

The hypothesis that lateral plate mesoderm cells would contribute to muscle formation (even though through specialized developmental channels) would be a major discovery in the field and clearly this is an exciting perspective. The approaches that have been taken in this study are original, combining avian work (including quail chick chimeras), mouse genetics, and scRNAseq. It should be pointed that illustrations are of very high quality and that there is a lot of work already placed in this study.

But given the importance that such findings would have, the evidence that are presented should be rock-solid. One of the problems with the identification of tiny subpopulations is that one wonders if they are real or they represent imperfections of the detection procedures that were used. Unfortunately, this work does not settle this question, as there are several issues with many of the performed experiments and a significant tendency to overinterpretation of the results that makes this study at best preliminary in the present form. Since this group has the technical abilities and the right

model to definitely settle the issue, I would encourage them to do the right experiments, whatever it takes.

General comment

1) The manuscript would gain in credibility if there were quantifications wherever appropriate. The phrases: we identified "a subpopulation", or we observed "a subset of cells" or "sparse cells" are way too vague at this level of publication. The authors should find a way to quantify their findings in an unbiased way.

Problems with the analyses of the quail-chick chimeras:

2) Quail chick chimeras are hard to perform cleanly and since the demonstration is dependent on the grafting skills of the experimenter, it is crucial to convince the reader that cells from the lateral plate/intermediate mesoderm were not carried in the presomitic grafts or cells from the presomitic mesoderm were not carried in the lateral plate grafts. Two schemes (Fig 1a and Fig 1h) are just not sufficient. Maybe grafting the medial half of the PSM or a lateral portion of the lateral plate mesoderm would avoid possible contaminations with cells originating from the border between the two tissues. A staining performed few hours after grafting would convince the reader of the cleanliness of the procedure. And since there were also questions on the invasiveness of quail-derived grafted tissues transplanted into chicken host, the way to address the issue of contamination could be done with the GFP chicken line that was also used here.

3) The QCPN antibody might be an easy tool to analyze quail chick chimeras, but it has its own limits. It was shown that it recognizes only 90-95% of all cells even in a bona fide quail tissue. This can be important when making assertive statements on whether a cell is or is not of quail origin in tiny subpopulations such as the ones that are described here.

4) In Fig 1 (presomitic grafts), the observation that nuclei derived from another tissue express myogenic markers after fusion to a myofiber is not surprising. Helen Blau has shown that in past decades (this is the theory of reprogramming in somatic cell heterokaryons). Such that this observation might in fact mean very little for the authors' demonstration, apart from the -very interesting- possibility that connective tissue cells fuse to muscle fibres.

5) In the same experiment, the observation that single cells express MyoG, but are not of quail origin is certainly more meaningful, but it is very surprising that there are so few of them (for instance on Fig 1f), and in fact there are just as many MyoG+ nuclei present in the muscle mass than in the midst of what seems to be the mesenchyme, which seems to be a very odd location. Could this mean that you are reaching the limit of this antibody's specificity and that it sometimes unspecifically picks up non-myogenic cells? Do the numbers of MyoG+ single cells of lateral plate origin that you observe fit with the scRNAseq numbers you performed? This has to be addressed.

Analyses of mice.

6) The conclusion that the tomato and the Pax7 do not colocalize is a little hasty, since the tomato is cytoplasmic, and Pax7 is nuclear. It could be that the Pax7-positive nuclei sometimes take so much space in the myofiber that there is very little cytoplasm and thus no tomato staining around them. The best way to be sure of a colocalization would be to have two nuclear markers. This has to be addressed.

7) "We observed sparse PAX7+ cells that were not Pax3-derived". If this theory is true, those sparse cells should be of SCX and Osr1 lineage, wouldn't they? Can this be proven? At the very least, the Pax7+ cells should be quantified in all three conditions and the numbers should fit.

scRNAseq

8) There are known artefacts of scRNAseq analyses (for instance encapsulated doublet cells) that could account for the tiny proportion of cells with double identity. What were the approaches that were used to avoid such problems?

BMP and Noggin over-expression

9) The authors interpret their data as a cell fate conversion (increase/decrease of Pax7 population "at the expense" of TCF4 population). Maybe this is true, but maybe not.

i) The argument that the observed differences cannot be due to proliferation because nuclei per unit area is unchanged does not really hold. Given the length of the experiment (about 5 days exposure to signals), and the average proliferation rate of cells in an embryo (about 10 hours), small differences in proliferation (maybe similar to those that are seen on the graphs) can lead to big differences after about twelve rounds of proliferation.

ii) Another point is that quantifications made on sections are very dangerous, since the shape and 3D occupancy of muscle bundles and tendons (and in fine, cell counts) are dependent on the level at which sections are made. Quantifications should be made differently.

iii) Authors have not considered alternative hypotheses, such as migration towards or away from signals which could account for the observed phenotype. One could also wonder whether the phenotypes observed is not due to the presence of a bolus of the RCAS-expressing cells that were injected.

Altogether, the interpretation of the data presented here is questionable and these experiments should be repeated using tools that can really monitor cell fate conversion (e.g. mouse genetics approaches or the grafting techniques that was used here).

10) In fact, if the general hypothesis put forward here of CT/muscle conversion were true, there would be an increasing number of lateral plate/CT derived nuclei incorporated in muscles during embryonic life. That should not be very difficult to prove using the techniques described here. This experiment would definitely settle most issues raised above and it is important that it is performed.

Reviewer #3 (Remarks to the Author):

The authors sought to define the mechanisms underlying formation of the muscle/tendon interface. To achieve this purpose, they took advantage of a wide array of approaches, such as quail-to-chicken xenograft experiments, fate mapping/lineage tracing experiments using Scx-Cre, Osr1-CreERT2 and scRNAseq analyses. They found that quail lateral plate mesoderm (LPM) contributes to muscles, and genetically marked fibroblasts become muscle cells near the interface. They also found in scRNAseq analysis that cells with fibroblast-myoblast dual identities are enriched for BMP activities, and manipulation of BMP signaling led to impaired formation of the interface. From these data, they conclude that a fibroblast-to-myoblast conversion at the interface is important for patterning limb muscles.

This manuscript challenges the classical model on the somatic mesoderm origin of myogenic cells at the muscle tips close to tendons. I have some concerns on the validity of the approaches and the interpretation of scRNAseq data. Importantly, definitive evidence for a fibroblast-to-myoblast conversion appears to be missing from the manuscript.

1. Can the authors comment on the validity of their very elegant quail-to-chick xenograft transplantation experiments? Are the transplants completely free of contaminating cells, or the recipient site completely free of pre-existing cells? I feel that we cannot completely exclude the

possibility that chicken myoblasts in quail presomitic grafts are due to pre-existing chicken somite, and quail myoblasts in quail lateral plate grafts are due to contaminating quail somite.

2. It is unclear whether Scx-Cre and Osr1-CreERT2 are truly specific to connective tissue fibroblasts. At least based on the data presented, Scx-GFP is widely expressed in the interface, and Scx-Cre appears to mark a number of cells without much specificity. Some forms of quantitative analyses are required to justify the use of these lines as CT-specific. Additionally, the authors did not show carefully the cells that are marked by Osr1-CreER shortly after tamoxifen injection. When were these mice treated with tamoxifen, and how many days afterward were they analyzed? Without these pieces of information, it is hardly possible to tell if these tools are specific to fibroblasts.

3. The authors' scRNAseq analyses clearly demonstrate that muscle cells and connective tissue cells are completely separate, without any cells with a dual CT/M identity at all stages examined. If such cells with dual CT/M identity exist, these cells should form their own clusters somewhere between the muscle and CT clusters in UMAP plot. Moreover, if a fibroblast-to-myoblast conversion indeed occurs, these cells should be recognized as transitional cell clusters between muscle and CT clusters. I would infer from this scRNAseq data that no fibroblast-to-myoblast conversion occurs at any stage.

Other points:

4. Line 115: The authors should quantify the percentage of non-Pax3-derived PAX7+ cells at two locations, in proximity to the interface and elsewhere, to strengthen this finding.

5. Line 141-144: This conversion model does not seem to be supported by the scRNAseq data shown below.

6. Line 170-171: Did the authors identify cells with the CT/M identity expressing both PRRX1 and TWIST2 in vivo around the interface?

7. Line 184-186: These pSMAD staining data at muscle tips are very convincing.

RESPONSE TO REVIEWER COMMENTS

We have reformatted the whole manuscript (that was initially a short version), to respond to the comments and suggestions of Reviewers 1, 2 and 3. We have extended the number of Figures from 4 to 8, slightly reduced the supplementary Figures (from 10 to 9) and included an extended discussion. Because we made extensive reformatting, the revision mark version was not very helpful.

Here is the list of the experimental changes we made:

- To respond to the concerns of Reviewers 2 and 3 about the quail-into-chicken grafting experiments, we have performed electroporation of lateral-plate mesoderm at the limb level with a nuclear reporter and confirmed the recruitment of lateral-plate derived cells to the myogenic lineage (Supplementary Fig. 4).
- As requested by Reviewers 2 and 3, we have quantified all that was technically possible to quantify. The numbers are self-consistent. Numbers have been added to the manuscript in Fig. 1h; Fig. 2d,f,h; Fig. 3f,m, Supplementary Fig. 5d,k and Supplementary Fig. 6f,j. We believe that the quantifications improve the manuscript.
- To answer Reviewer 2 concern about the colocalization of nuclear PAX7 labelling and cytoplasmic Tomato labelling, we have now performed *Pax3* lineage-tracing experiments using a nuclear GFP reporter (Fig. 2) and found the same results as with the cytoplasmic Tomato reporter, *i.e* the presence of PAX7+ and MYOD+ myogenic cells that were not *Pax3* lineage-derived at muscle tips.
- We have better analysed and explained the scRNAseq data to show that the dual-identity cells are in transit from a connective tissue state to a muscle state (Fig. 4, Supplementary Fig. 8) and as such, do not cluster independently from the connective tissue and muscle.
- To respond to Reviewer 3 on the timing of Tamoxifen induction in the *Osr1*CreERT2 lineage experiment, we performed induction at different time points (E10.5, E12.5, E13.5) (Fig. 3, Supplementary Fig. 6). They show that *Osr1*-fibroblasts are recruited to the muscle lineage before E13.

The reviewer comments are highlighted in grey and in italics. We have responded point by point to the reviewer comments.

Reviewer #1 (Remarks to the Author):

In this paper, the authors use both chick/quail grafting and mouse lineage tracing experiments to revisit the origin of limb muscle and connective tissue. The authors identify a novel contribution of the lateral plate to the forming musculature, mostly to the part that connects with the connective tissues. This finding was further supported by a single-cell analysis, where a population of cells that express a combination of myogenic and connective tissue genes was found. Finally, the authors suggest that BMP signaling regulates this conversion of fibroblasts to myoblasts.

We thank Reviewer1 for acknowledging the potential of our findings on the contribution

of the lateral plate mesoderm to the developing muscle in the CT/muscle interface.

In contrast to the part that describes the contribution of lateral plate mesoderm to the developing muscle in the CT/muscle interface, the part on the role of BMP regulation is not clear.

Overexpressed BMP activity in chicken limbs increased in the number of PAX7+ muscle cells and reduced TCF4+ cells. The authors state that “the increase of PAX7+ progenitors occurred at the expense of TCF4+ fibroblasts”. However, unless I missed something, they provide no direct evidence that TCF4+ cells convert to PAX7+ cells.

The BMP GOF and LOF *in vivo* experiments show an inverse correlation between the numbers of PAX7+ cells and TCF4+ fibroblasts with no change in total cell number. Because there is no change in total cell number, we conclude that upon BMP GOF experiments, the increase of PAX7+ muscle progenitors occurred at the expense of TCF4+ fibroblasts, while upon BMP LOF experiments the number of TCF4+ cells increase at the expense of PAX7+ cells. These *in vivo* BMP GOF and LOF experiments do not directly show a conversion between cell types, and we did not state this in the text. Instead, they suggest a conversion between cell types. The conversion is directly shown with the BMP GOF and LOF experiments in myoblast and fibroblast cultures. BMP inhibition transforms myoblasts into fibroblasts, while BMP has the ability to induce PAX7 expression in fibroblast cultures. The absence of any BMP4 effect on the number PAX7+ cells in myoblast cultures (in which there is no fibroblast to be converted) supports our proposal of a fibroblast-to-myoblast conversion upon BMP4 exposure in chicken limbs.

Inducing BMP overexpression in chick/quail embryos with grafted lateral plate could provide such evidence.

We fully agree with the reviewer that BMP overexpression in lateral plate quail/chick chimeras would provide the *in vivo* evidence of TCF4+ fibroblast conversion to PAX7+ muscle progenitors and that Noggin overexpression (BMP LOF) in somite quail/chick chimeras would provide the evidence of PAX7+ muscle progenitor conversion to TCF4+ fibroblasts. Unfortunately, these experiments are technically challenging, since the embryos would not support 2 successive surgical manipulations. For this reason, we turned to *in vitro* experiments to show the fibroblast-to-myoblast conversion.

Moreover, following this logic, the number of CT cells that can potentially differentiate into muscle cells is not restricted, implying that this phenomenon is not unique to a population that forms the CT/muscle interface.

Our results do not favour this suggestion made by the Reviewer. Indeed our *in vivo* and *in vitro* BMP functional experiments indicate that only a subset of CT cells have the ability to differentiate into myogenic cells. In BMP GOF experiments in limbs (*in vivo*), the increase of PAX7+ cells only occurs in muscle areas (including myogenic cells and associated connective tissue fibroblasts) leading to muscle fusion and patterning defects, but not in connective tissue close to the ectoderm despite the presence of ectopic BMP (see Figure 7a-d). Moreover, BMP overexpression in chicken embryonic fibroblast cultures only induces PAX7 expression in a subset of fibroblasts, suggesting that only a subset of fibroblasts have the ability to convert into myoblasts (see Figure 7j-m). We also believe that the fibroblast-to-muscle conversion is stage specific, since Tamoxifen

induction in mice after E12.5 leads to very few (Induction at E12.5) or no (Induction at E13.5) *Osr1*-derived myogenic cells (See Supplementary Fig. 6).

*I think that the authors should better explain their argument in the **Discussion**. Is this a unique population, or can any fibroblast transform into myoblast? What restricts this transformation in vivo? How is the BMP pathway involved in this activation/repression? These questions should be addressed.*

We thank the reviewer for raising these valid questions. We believe that only a subset of CT fibroblasts have this capacity is provided by BMP signalling that is restricted at the muscle/tendon interface during development. We have modified the text extensively to better explain these points in the discussion.

Finally, BMP4 was previously shown to regulate the differentiation of cells that connect between tendons and bones. It is important to discuss these findings and compare them to those of the current study. Overall, following the requested revisions, I support the publication of this manuscript.

We thank the reviewer for raising this point. BMP4 has indeed been shown to regulate the differentiation of cells at the bone/tendon interface. We have included this reference in the discussion in relation to the role of BMP at the tendon/muscle interface.

Reviewer #2 (Remarks to the Author):

In this paper, Esteves de Lima et al. examine the possibility that fibroblasts originating from the lateral plate mesoderm undergo a fibroblast to muscle conversion, and become integrated into growing myofibres during embryonic development. Their experiments indicate that this contribution is under the control of BMP signals.

*The hypothesis that lateral plate mesoderm cells would contribute to muscle formation (even though through specialized developmental channels) would be **a major discovery in the field and clearly this is an exciting perspective**. The approaches that have been taken in this study are original, combining avian work (including quail chick chimeras), mouse genetics, and scRNAseq. It should be pointed that illustrations are of very high quality and that there is a lot of work already placed in this study.*

But given the importance that such findings would have, the evidence that are presented should be rock-solid. One of the problems with the identification of tiny subpopulations is that one wonders if they are real or they represent imperfections of the detection procedures that were used. Unfortunately, this work does not settle this question, as there are several issues with many of the performed experiments and a significant tendency to overinterpretation of the results that makes this study at best preliminary in the present form. Since this group has the technical abilities and the right model to definitely settle the issue, I would encourage them to do the right experiments, whatever it takes.

We thank the Reviewer for the critical assessment of our work.

General comment

1) The manuscript would gain in credibility if there were quantifications wherever appropriate. The phrases: we identified “a subpopulation”, or we observed “a subset of cells” or “sparse cells” are way too vague at this level of publication. The authors should find a way to quantify their findings in an unbiased way.

As requested by the Reviewer, we have performed more extensive quantifications wherever appropriate and technically possible. In chicken embryos, we have quantified the proportion of MYOG+ nuclei that were QCPN-negative (not somite-derived) versus the total number of MYOG+ cells (percentage) in presomitic grafts (Fig. 1f-h). In mouse embryos, we have analysed the proportion of PAX7+ cells and MYOD+ cells that were not from *Pax3* lineage (Fig. 2, Supplementary Fig. 5a-d). We have quantified the number of myogenic cells (PAX7+ and MYOD+/MYOG+ cells) that were of *Scx* lineage (Fig. 3a-f, Supplementary Fig. 5g-k). We have also quantified the number of cells of *Pax7* lineage that were *Osr1*-derived after Tamoxifen induction at E10.5 and fixation at E15.5 (Fig. 3g-m) and E12.5 (Supplementary Fig. 6c-j). Quantifications show that 3 to 9 % of myogenic cells are not *Pax3*-derived, but rather derived from *Scx* or *Osr1* lineages.

Problems with the analyses of the quail-chick chimeras:

2) Quail chick chimeras are hard to perform cleanly and since the demonstration is dependent on the grafting skills of the experimenter, it is crucial to convince the reader that cells from the lateral plate/intermediate mesoderm were not carried in the presomitic grafts or cells from the presomitic mesoderm were not carried in the lateral plate grafts. Two schemes (Fig 1a and Fig 1h) are just not sufficient. Maybe grafting the medial half of the PSM or a lateral portion of the lateral plate mesoderm would avoid possible contaminations with cells originating from the

border between the two tissues. A staining performed few hours after grafting would convince the reader of the cleanliness of the procedure. And since there were also questions on the invasiveness of quail-derived grafted tissues transplanted into chicken host, the way to address the issue of contamination could be done with the GFP chicken line that was also used here.

We fully agree with the Reviewer that these experiments are dependent on the grafting skills of the experimenter. We were attentive to this point, and like to point out that the presence of non-somite-derived muscle cells, regionalized at muscle tips close to tendon has been observed by 3 different experimenters. Marie-Ange Bonnin (Supplementary Fig.1e-h), Laurent Yvernogeu (Fig.1a-h, Supplementary Fig.1a-d; Supplementary Fig. 2) who are both authors on the paper. Similar results were observed in presomitic grafts performed by Marie-Aimée Teillet, 20 years ago (see Figure below). I was personally amazed and reassured to recover this image/result dating to 20 years ago, from my postdoc in Nicole Le Douarin's lab.

[Redacted]

Given that 3 different experimenters over a 20 year period observed the same regionalized zone of non-somite-derived muscle cells at muscle tips, we feel that this provides a solid argument to exclude issues of grafting contamination. Moreover, we have now performed additional lateral-plate mesoderm electroporation at the limb level with a fluorescent nuclear reporter (nuclear GFP) that becomes stably integrated into the chicken genome in electroporated cells (Supplementary Fig. 4). Despite the fact that the electroporation technique does not target all cells, we did observe the same phenotype than with the grafts, *i.e* sporadic myonuclei, MYOG+ and PAX7+ myogenic cells that were GFP+ (lateral-plate derived), regionalized at muscle tips close to tendons (Supplementary Fig. 4). As expected, we also found pSMAD1/5/9+ cells that were GFP+ in the same region (Supplementary Fig. 4). This result shows the contribution of lateral-plate mesoderm derived cells to myogenic lineage with an independent technique, other than quail-into-chicken grafting experiments.

We are now even more convinced that avian chimeras and lateral-plate electroporation together show the existence of a pool of myogenic cells of lateral-plate origin localized at

muscle tips in chicken. This unexpected contribution of CT fibroblasts to muscle cells is further confirmed with genetic lineage tracing analysis in mice during foetal development (*Pax3-Cre*, *Scx-Cre* and *Osr1-Cre*) (Figs 2 and 3, Supplementary Figs. 5 and 6), and also at postnatal stage (*Prrx1-Cre*) (Yaseen Badarneh et al., back-to-back submission)

It is crucial to convince the reader that cells from the lateral plate/intermediate mesoderm were not carried in the presomitic grafts

We thank the reviewer for raising this concern. Yet, if ectopic quail lateral-plate mesoderm were carried out with quail presomitic grafts, we would see quail tendon cells or quail connective-tissue cells, which is never the case, showing that we had no such contamination during the grafting experiments.

3) The QCPN antibody might be an easy tool to analyze quail chick chimeras, but it has its own limits. It was shown that it recognizes only 90-95% of all cells even in a bona fide quail tissue. This can be important when making assertive statements on whether a cell is or is not of quail origin in tiny subpopulations such as the ones that are described here.

In our hands, the QCPN antibody labels all nuclei on quail sections and in quail cell cultures. Further, the QCPN antibody reliability would be an issue if the QCPN-negative quail cells were evenly distributed within muscle following quail-into-chicken presomitic grafts. This is not the case as the QCPN-negative cells that we observed are regionalized within muscles. It seems highly unlikely that the undetected quail cells would be specifically localized at muscle tips. Moreover, we have similar results with GFPchicken-into-chicken presomitic grafts, for which we do not use the QCPN antibody (Supplementary Fig. 2). Lastly, quail-into-chicken lateral plate grafts and lateral plate electroporation with a nuclear GFP reporter provide the same results that are the mirror-image of the presomitic grafts (Fig. 1i-l; Supplementary Fig. 4). Taken together, we believe that we provide strong evidence that the identification of lateral plate-derived muscle cells does not rely on the lack of reliability of the QCPN antibody.

4) In Fig 1 (presomitic grafts), the observation that nuclei derived from another tissue express myogenic markers after fusion to a myofiber is not surprising. Helen Blau has shown that in past decades (this is the theory of reprogramming in somatic cell heterokaryons). Such that this observation might in fact mean very little for the authors' demonstration, apart from the -very interesting- possibility that connective tissue cells fuse to muscle fibres.

We thank the reviewer for raising this interesting analogy. While we document CT cells that are incorporated to muscle fibres, it is not clear whether CT fibroblasts acquire a full myogenic identity before or after fusion during foetal myogenesis. We observe non-somite-derived cells that express muscle markers before fusion, such as mononucleated MYOG+ myoblasts (Fig. 1f-h) and PAX7+ muscle progenitors (Fig. 3d-j, m, Supplementary Fig. 6c-f). This thus suggests that the onset of reprogramming occurs before fusion during foetal myogenesis. We now discuss this point in the discussion. The reprogramming onset before fusion is also observed in postnatal myogenesis (Yaseen Badarneh et al., back-to-back submission)

5) In the same experiment, the observation that single cells express MyoG, but are not of quail

origin is certainly more meaningful, but it is very surprising that there are so few of them (for instance on Fig 1f), and in fact there are just as many MyoG+ nuclei present in the muscle mass than in the midst of what seems to be the mesenchyme, which seems to be a very odd location. Could this mean that you are reaching the limit of this antibody's specificity and that it sometimes unspecifically picks up non-myogenic cells?

The Reviewer is correct to point out that sparse MYOG+ cells are observed outside individual muscles. This has been observed in other studies and by other groups. This labelling is not due to unspecific labelling with the MYOG antibody as those outsider cells express MYOG transcripts as well. These outsider MYOG + cells are QCPN negative. For now, we do not know the significance of these outsider MYOG + cells.

Do the numbers of MyoG+ single cells of lateral plate origin that you observe fit with the scRNAseq numbers you performed? This has to be addressed.

We thank the Reviewer for pointing this out. We have now quantified the non-somite-derived MYOG+ cells and found that they represent 9% of the MYOG+ myoblasts at E10 in our somite grafting experiments (Fig. 1h). In the scRNAseq analysis at E10, MYOG+ dual cells represent 11% of the MYOG+ myoblasts. Therefore, the numbers are consistent and this has been added to the revised MS.

Analyses of mice.

6) The conclusion that the tomato and the Pax7 do not colocalize is a little hasty, since the tomato is cytoplasmic, and Pax7 is nuclear. It could be that the Pax7-positive nuclei sometimes take so much space in the myofiber that there is very little cytoplasm and thus no tomato staining around them. The best way to be sure of a colocalization would be to have two nuclear markers. This has to be addressed.

We initially used the Tomato reporter line (*Madisen et al., 2010, PMID: 20023653*), which is expressed in cytoplasm and nuclei (Supplementary Fig. 5a-c), and we detected PAX7+ cells with no Tomato (neither in the nucleus nor in the cytoplasm) in limb muscles of E15.5 (Fig. 2a-d) and E14.5 (Supplementary Fig. 5a-d) Pax3Cre:Rosa26Tomato embryos. As suggested by Reviewer2, we repeated the experiments using a nuclear reporter line (H2B-GFP) and confirmed the presence of PAX7+ and MYOD+ myogenic cells that were GFP-negative at muscle tips in E15.5 Pax3Cre:H2B-GFP embryos (Fig. 2e-h). These results are now added to the manuscript.

7) “We observed sparse PAX7+ cells that were not Pax3-derived”. If this theory is true, those sparse cells should be of SCX and *Osr1* lineage, wouldn't they? Can this be proven? At the very least, the Pax7+ cells should be quantified in all three conditions and the numbers should fit.

We agree that this point can be addressed genetically, however it implies a significant amount of mouse work that is prohibitive in the current pandemic-restrictive conditions.

Nevertheless, we quantified and compared the numbers that we obtained with the different reporter lines. These data are now included in the revised text and Figures.

At E14.5, we found 7.5% of PAX7+ cells that were not *Pax3*-derived using the Pax3Cre:R26Tom line (Supplementary Fig. 5a-d). Conversely, we found 7.4% of PAX7+ cells that were *Scx*-derived at E14.5 (Fig. 3f) using the ScxCre:R26Tom:Scx-GFP line. These numbers fit perfectly.

At E15.5, we quantified 4.6% (Tomato/E15.5) and 3.1% (GFP/E15.5) of PAX7+ that were not *Pax3*-derived. Conversely, we found 4.5% of betaGal+ cells (PAX7) that were *Osr1*-derived at E15.5 (Fig. 3m) after tamoxifen injection at E11.5. Although they are small numbers and there are small differences between reporters, the overall numbers do fit again.

scRNAseq

8) There are known artefacts of scRNAseq analyses (for instance encapsulated doublet cells) that could account for the tiny proportion of cells with double identity. What were the approaches that were used to avoid such problems?

We thank Reviewer2 for his/her important comment. We were aware of this possibility, that is why potential doublets were identified by running the Scrublet algorithm (*Wolock, S. L., Lopez, R. & Klein, A. M. Scrublet: Computational Identification of Cell Doublets in Single-Cell Transcriptomic Data. Cell Syst 8, 281-291.e9 (2019) PMID: 30954476*) and then removed from the datasets. This is described in the Materials and Methods.

BMP and Noggin over-expression

9) *The authors interpret their data as a cell fate conversion (increase/decrease of Pax7 population “at the expense” of TCF4 population). Maybe this is true, but maybe not.*

i) The argument that the observed differences cannot be due to proliferation because nuclei per unit area is unchanged does not really hold. Given the length of the experiment (about 5 days exposure to signals), and the average proliferation rate of cells in an embryo (about 10 hours), small differences in proliferation (maybe similar to those that are seen on the graphs) can lead to big differences after about twelve rounds of proliferation.

We thank the reviewer for allowing us to clarify this point. As the reviewer says, because there are 5 days of exposure to signals (BMP or Noggin), small changes in cell proliferation occurring every 10 hours should lead to big differences in total cell number at E10 (DAPI staining). As there is clearly no change in total cell number in muscle (myogenic cells and CT fibroblasts) in the 3 conditions (control, BMP and NOGGIN) at E10 after 5 days of exposure (Fig. 7g,h; Fig. 8g,h), we are confident that cell proliferation is not overtly affected in limb muscles in these BMP GOF and LOF experiments. Moreover, the EdU experiments performed at E9 on a short interval (6 hours of incubation of EdU) (Supplementary Fig. 9a-g) agree with the absence of change in the proliferation rate of PAX7+ cells in BMP GOF experiments. The absence of change in cell proliferation is also supported with the BMP GOF experiments in myoblast cultures (Supplementary Fig. 9h-j).

ii) Another point is that quantifications made on sections are very dangerous, since the shape and 3D occupancy of muscle bundles and tendons (and in fine, cell counts) are dependent on the level at which sections are made. Quantifications should be made differently.

This is a fair point and we are definitely aware of this. We were (and are always) very careful to count cells in muscles from limb sections obtained at the same level of the proximo-distal axis (see examples of control and experimental limbs sectioned at the same proximo-distal level, at low magnification (Fig. 7c,d, Fig. 8c,d).

iii) Authors have not considered alternative hypotheses, such as migration towards or away from signals which could account for the observed phenotype. One could also wonder whether the phenotypes observed is not due to the presence of a bolus of the RCAS-expressing cells that were injected.

We do not formally exclude a migration hypothesis. However, since we observe an inverse correlation between PAX7+ muscle progenitors and TCF4+ CT fibroblasts in both BMP GOF and LOF experiments with no change in total cell number, we favour the recruitment hypothesis, which is also supported by the myoblast and fibroblast culture experiments (Fig. 7,8, supplementary Fig. 9). We have previously checked that there is no bolus of RCAS-expressing cells (*Duprez et al., 1996 PMID: 8843392*). The extent of RCAS virus spread in dorsal limb regions with BMP4/RCAS or NOGGIN/RCAS (Fig. 7b, Fig. 8b) is not consistent with the presence of a bolus of cells but rather with the graft position in dorsal limb regions.

Altogether, the interpretation of the data presented here is questionable and these experiments should be repeated using tools that can really monitor cell fate conversion (e.g. mouse genetics approaches or the grafting techniques that was used here).

We fully agree that BMP overexpression in lateral-plate quail/chick chimeras would provide direct evidence of TCF4+ fibroblast conversion to PAX7+ muscle progenitors and that Noggin overexpression (BMP LOF) in somite quail/chick chimeras would provide the evidence of PAX7+ muscle progenitor conversion to TCF4+ fibroblasts. However, these experiments are technically extremely challenging, since the embryos would have to bear 2 successive surgical manipulations. This is why we turned to *in vitro* myoblast and fibroblast experiments to show a conversion phenotype.

10) In fact, if the general hypothesis put forward here of CT/muscle conversion were true, there would be an increasing number of lateral plate/CT derived nuclei incorporated in muscles during embryonic life. That should not be very difficult to prove using the techniques described here. This experiment would definitely settle most issues raised above and it is important that it is performed.

Our hypothesis is not that there is a constant flux of CT/M cells during development but rather that the majority of CT-to-muscle conversion events would occur around E6 in chicken (E12.5 in mouse embryos), at the stage where spatial arrangements occur to pattern limb muscles. We did not observe many myogenic cells of *Osr1* lineage upon tamoxifen induction after E12.5 (Supplementary Fig. 6c-j). Consistently, the number of dual cells versus all muscle cells drops from 3-9% at foetal stages (our *in vivo* study) to 0.05% in postnatal muscle (*Yaseen Badarneh et al., back-to-back submission*). Our hypothesis is that the CT-to-muscle conversion occurs mainly at the crucial stages of spatial muscle and CT rearrangements.

Reviewer #3 (Remarks to the Author):

The authors sought to define the mechanisms underlying formation of the muscle/tendon interface. To achieve this purpose, they took advantage of a wide array of approaches, such as quail-to-chicken xenograft experiments, fate mapping/lineage tracing experiments using Scx-Cre, Osr1-CreERT2 and scRNAseq analyses. They found that quail lateral plate mesoderm (LPM) contributes to muscles, and genetically marked fibroblasts become muscle cells near the interface. They also found in scRNAseq analysis that cells with fibroblast-myoblast dual identities are enriched for BMP activities, and manipulation of BMP signaling led to impaired formation of the interface. From these data, they conclude that a fibroblast-to-myoblast conversion at the interface is important for patterning limb muscles.

This manuscript challenges the classical model on the somatic mesoderm origin of myogenic cells at the muscle tips close to tendons. I have some concerns on the validity of the approaches and the interpretation of scRNAseq data. Importantly, definitive evidence for a fibroblast-to-myoblast conversion appears to be missing from the manuscript.

1. Can the authors comment on the validity of their very elegant quail-to-chick xenograft transplantation experiments? Are the transplants completely free of contaminating cells, or the recipient site completely free of pre-existing cells? I feel that we cannot completely exclude the possibility that chicken myoblasts in quail presomitic grafts are due to pre-existing chicken somite, and quail myoblasts in quail lateral plate grafts are due to contaminating quail somite. We agree that we cannot completely exclude this possibility. However, in presomitic mesoderm grafts (quail into chicken), a contamination of pre-existing chicken presomitic mesoderm cells would rather lead to chicken cells distributed in a non-regionalised manner within muscles. The concentration of chicken muscle cells at muscle tips argues against a simple contamination issue. Moreover, the presomitic mesoderm grafts have been performed by 3 different experimenters and provide the same results (see response to Reviewer1) - the presence of non-somite-derived cells at muscle tips close to tendons. The fact that the lateral-plate grafts (quail into chicken) give the converse result, *i.e.* regionalised lateral-plate derived cells at muscle tips, is another indirect argument against contamination. Nevertheless, in order to bypass the potential contamination problem of lateral-plate grafts, we performed lateral plate electroporation with a nuclear GFP reporter gene. Electroporation only targets a fraction of lateral-plate cells. Nevertheless, we observed GFP+ nuclei in muscle cells at muscle tips (GFP+ myonuclei, GFP+/MYOG+ cells; GFP+/PAX7+ cells and GFP+/pSMAD1/5/9+ myonuclei) (Supplementary Fig. 4). We are quite confident that a contamination issue cannot explain our observations. The fact that mouse genetic lineage experiments give similar results is an additional argument supporting the results of quail-into-chicken grafts.

2. It is unclear whether Scx-Cre and Osr1-CreERT2 are truly specific to connective tissue fibroblasts. At least based on the data presented, Scx-GFP is widely expressed in the interface, and Scx-Cre appears to mark a number of cells without much specificity. Some forms of quantitative analyses are required to justify the use of these lines as CT-specific. Additionally, the authors did not show carefully the cells that are marked by Osr1-CreER shortly after tamoxifen

injection. When were these mice treated with tamoxifen, and how many days afterward were they analyzed? Without these pieces of information, it is hardly possible to tell if these tools are specific to fibroblasts.

Both *Scx* and *Osr1* are recognized markers for two types of connective tissue fibroblasts. *Scx* labels tendons, a regular connective tissue (Murchison et al., 2007, PMID: 17567668, Schweitzer et al., 2001, PMID: 11585810). As previously described, *Scx:Cre* labels limb tendons (Fig. 3a). *Osr1* labels muscle irregular connective tissue and drives the differentiation program of irregular connective tissue (Vallecillo et al., 2017, PMID: 29084951). As previously described, *Osr1:Cre* labels limb irregular connective tissue (Supplementary Fig. 6a,b).

Scx:Cre:

The muscle cells of *Scx* lineage are concentrated at the muscle/tendon interface at E12.5 (Supplementary Fig. 5g-j) and close to tendon at E14.5 (Fig. 3a-e). The quantification of PAX7+ cells and MYOD+/MYOG+ cells that are *Scx*-derived are respectively 7.38 % and 4.8% at E14.5 (Fig.3f), 4.12% and 4.13% at E12.5 (Supplementary Fig. 5k). These numbers are consistent with the percentage of PAX7+ cells that are not *Pax3*-derived at E15.5 (3 to 5%, Fig. 2d,f) and at E14.5 (7.5%, Supplementary Fig. 5d).

Osr1:CreERT2:

Tamoxifen injection at E10.5 leads to 4.5% of PAX7+ cells and 3.6% of MYOD+/MYOG+ cells that are of *Osr1* lineage at E15.5 and to 6.25 % of PAX7+ cells and 6.9% of MYOD+/MYOG+ cells that are of *Osr1* lineage at E12.5. Again, these numbers are consistent with the percentage of PAX7+ cells that are not *Pax3*-derived at E15.5 (3 to 5%, Fig. 2d,f) and at E14.5 (7.5%, Supplementary Fig. 5d).

The consistency of the above numbers strongly suggests that the *Scx:Cre* and *Osr1:CreERT2* mouse lines do not introduce biases in our analysis.

As requested, we performed a short-term Tamoxifen induction experiment (induction at E10.5 – fixation at E12.5). The size of the *Osr1* lineage-derived myogenic populations is slightly larger than for the E15.5 fixation (see numbers above), which is consistent with the hypothesis that the fibroblast recruitment peaks around E12.5.

As requested, we clarified the induction and fixation stages for the Tamoxifen induction experiments in the manuscript.

These data are now clearly mentioned in our revised version of the manuscript.

3. The authors' scRNAseq analyses clearly demonstrate that muscle cells and connective tissue cells are completely separate, without any cells with a dual CT/M identity at all stages examined. If such cells with dual CT/M identity exist, these cells should form their own clusters somewhere between the muscle and CT clusters in UMAP plot. Moreover, if a fibroblast-to-myoblast conversion indeed occurs, these cells should be recognized as transitional cell clusters between muscle and CT clusters. I would infer from this scRNAseq data that no fibroblast-to-myoblast conversion occurs at any stage.

We thank the reviewer for allowing us to clarify this point. A population in transit between two states would not be identified as a continuum between two states if the switch between the two states is achieved quickly via down-regulation of state 1 genes and up-regulation of state 2 genes. As this population shifts its expression profile from state 1 to

state 2, it is expected to cluster with population 1 around the shift onset and with population 2 towards the end of the shift. This is indeed what we observe (Fig. 4c-e; Supplementary Fig.8b; Supplementary Table 2). This population in transit would only cluster if in addition to exhibiting intermediate profiles between states, it would express specific genes, which is not the case here. It is the case in the Yaseen Badarneh et al. study (back-to-back submission) where dual cells harbour a myotendinous junction-specific expression profile, in addition to being observed in CT and muscle clusters. We believe that the difference is due to the fact that in our study, it is too early in development for the dual cells to exhibit a myotendinous junction specific signature. We have modified the Result section accordingly to make this point clearer.

To rule out the possibility that the dual cells correspond to encapsulated doublet cells, potential doublets were identified by running the Scrublet algorithm (Wolock, S. L., Lopez, R. & Klein, A. M. *Scrublet: Computational Identification of Cell Doublets in Single-Cell Transcriptomic Data. Cell Syst* **8**, 281-291.e9 (2019) PMID: 30954476) and then removed from the datasets. This is described in the Materials and Methods.

Other points:

4. Line 115: The authors should quantify the percentage of non-Pax3-derived PAX7+ cells at two locations, in proximity to the interface and elsewhere, to strengthen this finding.

We have now quantified the percentage of non-*Pax3*-derived PAX7+ cells and found that 7.5% of PAX7+ cells at E14.5 (Supplementary Fig. 5d) and 4.6% of PAX7+ cells at E15.5 (Fig. 2d) were not *Pax3*-derived. These cells were preferentially observed at the muscle/tendon interface. Although labelling was preferentially restricted to muscle tips, we counted all the muscle cells present in the field of view to compare with the numbers derived from the scRNAseq analysis and because it was not possible to define the border between the central zone and the tips in a reproducible manner. This comment has been added to the Material and Methods.

5. Line 141-144: This conversion model does not seem to be supported by the scRNAseq data shown below.

The identification of a small population of limb cells expressing both muscle and CT makers in the same cell (even if not clustered) shows the existence of cells with a dual identity. Because these dual cells express an intermediate expression level of muscle and CT markers compared to strict muscle and CT cells (Fig. 4c), this is suggestive of a transitional state between CT and muscle identity.

6. Line 170-171: Did the authors identify cells with the CT/M identity expressing both PRRX1 and TWIST2 in vivo around the interface?

We did not perform this double in situ experiments as we were limited in the types of experiments to prioritise due to the ongoing pandemic

7. Line 184-186: These pSMAD staining data at muscle tips are very convincing.

This very convincing regionalised pSMAD staining supports a specific function for myonuclei with active BMP signalling at muscle tips close to tendons.

REVIEWERS' COMMENTS

Reviewer #1 (Remarks to the Author):

The author addressed all my concerns, I support its publication

Reviewer #2 (Remarks to the Author):

I believe the authors have addressed the comments that were made appropriately. They made a number of experiments, quantifications and text changes that have made the manuscript stronger. Good work and well done

Reviewer #3 (Remarks to the Author):

Regarding the three major points that I raised previously,

1. The authors strengthened the conclusion from the quail-to-chick xenograft transplantation study by performing an additional lateral plate electroporation experiment enabling lineage tracing. The response is acceptable.
2. The authors justified Scx:Cre and Osr1:CreERT2 lines as CT-specific by performing additional quantification and pulse-chase experiments. The response is acceptable.
3. Regarding dual CT/M identity cells not showing as a separate cluster in their scRNAseq data, the authors' explanation is puzzling, particularly in light of the findings from the companion manuscript under consideration. However, as this point is not central to supporting their conclusion, this reviewer finds this response acceptable.

It is somewhat disappointing that the authors did not take measures to validate their scRNAseq findings during the revision, by means of in situ experiments for PRRX1 and TWIST2, as they did not agree with the priority of such experiments.

Overall, the authors present a new concept in this study through a number of technically sound experiments.